# SRF-deficient astrocytes provide neuroprotection in mouse models of excitotoxicity and neurodegeneration

Surya Chandra Rao Thumu[1], Monika Jain[1], Sumitha Soman[1], Soumen Das[1], Vijaya Verma[2], Arnab Nandi[1], David H Gutmann[3], Balaji Jayaprakash[1], Deepak Nair[1], James P Clement[2†], Swananda Marathe[1‡], Narendrakumar Ramanan[1]*

[1]Centre for Neuroscience, Indian Institute of Science, Bangalore, India; [2]Neuroscience Unit, Jawaharlal Nehru Centre for Advanced Scientific Research, Bangalore, India; [3]Department of Neurology, Washington University School of Medicine, St. Louis, United States

**Abstract** Reactive astrogliosis is a common pathological hallmark of CNS injury, infection, and neurodegeneration, where reactive astrocytes can be protective or detrimental to normal brain functions. Currently, the mechanisms regulating neuroprotective astrocytes and the extent of neuro-protection are poorly understood. Here, we report that conditional deletion of serum response factor (SRF) in adult astrocytes causes reactive-like hypertrophic astrocytes throughout the mouse brain. These *Srf*GFAP-ERCKO astrocytes do not affect neuron survival, synapse numbers, synaptic plasticity or learning and memory. However, the brains of *Srf* knockout mice exhibited neuroprotection against kainic-acid induced excitotoxic cell death. Relevant to human neurodegenerative diseases, *Srf*GFAP-ERCKO astrocytes abrogate nigral dopaminergic neuron death and reduce β-amyloid plaques in mouse models of Parkinson's and Alzheimer's disease, respectively. Taken together, these findings establish SRF as a key molecular switch for the generation of reactive astrocytes with neuroprotective functions that attenuate neuronal injury in the setting of neurodegenerative diseases.

*For correspondence:
naren@iisc.ac.in

Present address: †University of Exeter Medical School, Hatherly Building, Exeter, United Kingdom; ‡Department of Biosciences and Bioengineering, Indian Institute of Technology, Dharwad, India

Competing interest: The authors declare that no competing interests exist.

## Editor's evaluation

This is an important study that uses a variety of complementary approaches to demonstrate that adult astrocytes lacking serum response factor are neuroprotective. The evidence supporting this conclusion is solid, reflecting mostly high quality cellular and molecular data with minor remaining concerns regarding the behavioral data.

## Introduction

CNS injury can be caused by diverse etiologies to result in either acute tissue damage (ischemia, traumatic brain injury) or chronic morphological and functional changes in neural tissues, resulting in behavioral and cognitive deficits as seen in neurodegenerative diseases (*Burda and Sofroniew, 2014*; *Pekny and Pekna, 2014*). Although much of the focus has centered on neuronal dysfunction, emerging studies have highlighted the critical roles played by non-neuronal cells, particularly astrocytes, in tissue repair, homeostasis, and disease progression (*Burda et al., 2016*; *Kimelberg, 2010*; *Linnerbauer and Rothhammer, 2020*; *Pekny and Pekna, 2014*). In this regard, astrocytes respond to CNS pathology by undergoing a spectrum of transcriptomal, physiological and structural changes, termed 'reactive astrogliosis or reactive astrocytosis' (*Burda and Sofroniew, 2014*;

*Liddelow and Barres, 2017*). Astrocyte reactivity is a diverse and complex cellular response that is context-dependent and reactive astrocytes perform several critical functions including aiding in repair and restoring normal homeostasis in the brain (*Aswendt et al., 2022*; *Li et al., 2008*; *Linnerbauer and Rothhammer, 2020*; *Pekny and Pekna, 2014*).

Although reactive astrocytes could provide neuroprotection in the initial stages of disease, prolonged gliosis could hamper normal neuronal functions and contributes to the pathophysiology of the disease (*Gleichman and Carmichael, 2020*; *Huang et al., 2022*; *Pekny and Pekna, 2014*; *Phatnani and Maniatis, 2015*; *Burda and Sofroniew, 2014*; *Verkhratsky et al., 2016*). For example, reactive astrocytes generated by lipopolysaccharide (LPS)-induced neuroinflammation are dependent on microglia, deficient in several critical astrocyte functions and cause death of neurons and oligodendrocytes (*Guttenplan et al., 2021*; *Liddelow et al., 2017*). Similarly, reactive astrocytes are also found in the aging brain and in the context of CNS neurodegeneration (*Boisvert et al., 2018*; *Clarke et al., 2018*; *Liddelow et al., 2017*), where inhibition of reactive astrocytes provides neuroprotection in mouse models of Parkinson's disease, Alzheimer's disease, and ALS (*Ceyzériat et al., 2018*; *Guttenplan et al., 2020b*; *Park et al., 2021*; *Reichenbach et al., 2019*; *Yun et al., 2018*). For these reasons, suppression of reactive astrogliosis is actively being pursued as an astrocyte-targeted therapeutic strategy for the treatment of neurodegenerative diseases (*Lee et al., 2022*).

Previous studies have identified several genes including *Stat3*, *Fgfr*, *Endothelin-1*, *β1-integrin* and *Bmal* whose deletion results in hypertrophic reactive-like astrocytes (*Correa-Cerro and Mandell, 2007*; *Kang and Hébert, 2011*; *Lananna et al., 2018*; *Sofroniew, 2014*). Furthermore, genetic manipulations of some of these genes have revealed their importance in regulating both the beneficial and negative effects of reactive astrocytes. For example, astrocyte-specific deletion of *Stat3* has revealed critical roles played by this pathway in the generation of scar-border forming astrocytes and tissue repair, and in the pathogenesis of Alzheimer's and Huntington diseases (*Abjean et al., 2023*; *Ben Haim et al., 2015*; *Herrmann et al., 2008*; *Okada et al., 2006*; *Reichenbach et al., 2019*). Deletion of β1-integrin resulted in progressive astrogliosis and spontaneous seizures in adult mice (*Robel et al., 2015*; *Robel et al., 2009*). These observations raise the intriguing possibility that reactive astrocytes can be reprogrammed to improve neuronal survival and promote CNS repair and recovery following injury and in the setting of neurodegenerative diseases (*Lee et al., 2022*).

To identify potential mechanisms for reactive astrocyte reprogramming, we focused on SRF, a stimulus-dependent transcription factor that plays several critical roles in nervous system development and glial differentiation (*Knöll et al., 2006*; *Knöll and Nordheim, 2009*; *Lu and Ramanan, 2011*; *Lu and Ramanan, 2012*). We recently showed that astrocyte-specific deletion of SRF early during mouse development resulted in persistent reactive-like astrocytes throughout the postnatal mouse brain (*Jain et al., 2021*). Although these astrocytes did not cause any discernible abnormalities in the brain, the phenotypic changes exhibited by these *Srf*-deficient astrocytes could be due to developmental defects in astrocyte differentiation caused by SRF deletion (*Jain et al., 2021*; *Lu and Ramanan, 2012*). To address this, we now report that deletion of SRF in adult astrocytes also causes astrocyte reactivity-like phenotype that is persistent and widespread across the brain. We further show that *Srf*-deficient astrocytes did not affect neuronal survival, normal neuron functions, or learning and memory. Importantly, we demonstrate that astrocytic *Srf* deletion results in markedly attenuated neuronal death caused by excitotoxicity, and in a mouse model of Parkinson's disease. Furthermore, astrocytic *Srf* deletion in the APP/PS1 mouse model of Alzheimer's disease causes a significant decrease in β-amyloid plaque burden. Taken together, our results reveal SRF as a critical regulator of neuroprotective reactive astrogliosis in the context of brain injury and neurodegenerative disease.

## Results

### SRF deletion in adult astrocytes results in GFAP+ hypertrophic astrocytes

We recently showed that astrocyte-specific SRF deletion during embryonic development using a GFAP-Cre transgenic mouse line (*Srf*<sup>GFAP</sup>CKO) results in reactive-like astrocytes across the brain starting around 2 weeks of age and that these astrocytes persist throughout adulthood (*Jain et al., 2021*). These changes in astrocytes could reflect a developmental effect or result from indirect effects via other cell types with Cre-mediated *Srf* loss (e.g. neurons). To determine whether SRF functions in

a cell-autonomous fashion in adult astrocytes to establish a non-reactive state, we generated $Srf^{f/f;}$ $^{GFAP-ERT+/-}$ ($Srf^{GFAP-ER}$CKO) mice in which the hGFAP promoter drives the expression of a tamoxifen-inducible Cre recombinase in postnatal astrocytes, rather than in GFAP-positive neural progenitor cells (*Figure 1—figure supplement 1*; *Chow et al., 2008*). In this hGFAP-Cre$^{ERT}$ transgenic line, co-staining for β-gal from the Cre-IRES-β-gal transgene and cell-type specific marker genes revealed that ~85% of GFAP +and Sox9 +astrocytes were β-gal-positive in the neocortex, corpus callosum and hippocampus (*Figure 1—figure supplement 1*). No Olig2+/β-gal+oligodendrocyte lineage cells and Iba1+/ β-gal+microglia were found (*Figure 1—figure supplement 1*), while fewer than 1% of NeuN + cells were β-gal+in the neocortex and striatum, and none were β-gal+in the hippocampus (*Figure 1—figure supplement 1*). To delete *Srf* in adult astrocytes, tamoxifen was administered to 6- to 8-week-old *Srf*$^{GFAP-ER}$CKO mice and control littermates after astrocyte development was complete (*Figure 1A*; *Wang and Bordey, 2008*).

We first confirmed SRF deletion in the astrocytes in the *Srf*$^{GFAP-ER}$CKO mice at 2 mpi by co-immunostaining for SRF and the astrocytic marker, S100β. The astrocytes in control mice showed robust expression of SRF while many astrocytes in the knockout mice did not show any SRF expression (*Figure 1B*). Co-immunostaining for Sox9 and SRF revealed that around 52–54% of astrocytes were deleted for astrocytes in the neocortex and hippocampus similar to that reported in this Cre driver line (*Chow et al., 2008*; *Figure 1C*). We observed that the *Srf*-deficient astrocytes exhibited increased branching with hypertrophic morphology as seen with S100β staining (*Figure 1D, E*). Immunostaining for the astrocytic marker, GFAP, showed little to no expression in the neocortex and striatum and only basal expression in the hippocampal astrocytes of 2-month-old tamoxifen injected control mice. In contrast, the astrocytes in the tamoxifen-injected mutant mice exhibited robust GFAP expression in all the brain regions analyzed (*Figure 1F, G*). Importantly, the associated phenotypic changes in astrocytes upon *Srf* deletion were not spatially restricted in the brain, as observed in studies using other knockout strains (*Garcia et al., 2010*; *Kang et al., 2014*). We next asked whether the increased GFAP expression and hypertrophic morphology of astrocytes was a transient phenomenon in the *Srf*$^{GFAP-ER}$CKO mice or persisted throughout adulthood. While there were few to no GFAP-positive astrocytes in the neocortex and striatum, and weakly GFAP-positive astrocytes in the hippocampus of control mice at 12 mpi, intense GFAP-positive hypertrophic astrocytes were found in several regions of the brain, including the neocortex, hippocampus, and striatum of *Srf*$^{GFAP-ER}$CKO mice (*Figure 1H, I*; *Figure 1—figure supplement 2*). These findings reveal that *Srf* deletion in adult astrocytes also causes widespread and persistent GFAP +hypertrophic astrocytes like that observed with embryonic deletion in astrocytes (*Jain et al., 2021*).

## *Srf*$^{GFAP-ER}$CKO mice show upregulation of reactive astrocyte markers

Hypertrophic morphology with increased GFAP expression is generally exhibited by reactive astrocytes, which show a heterogeneous context-dependent transcriptomic profile reflective of their diverse reactive states (*Das et al., 2020*; *Jiwaji et al., 2022*; *Zamanian et al., 2012*). To study whether the *Srf*-deficient astrocytes exhibit reactivity, we performed immunostaining for the reactive astrocyte marker, vimentin along with GFAP (*Ridet et al., 1997*). We found that the GFAP-positive astrocytes in *Srf* cKO mice were also vimentin-positive, whereas no vimentin+/GFAP +astrocytes were found in the control mice (*Figure 2A, B*). We next analyzed the expression of genes that were described for reactive astrocytes induced in response to two acute pathological paradigms – lipopolysaccharide (LPS)-induced neuroinflammation and ischemic stroke (*Liddelow et al., 2017*; *Zamanian et al., 2012*). Expression levels were determined by real-time qRT-PCR of cortical tissue for the following genes: *Ugt1a1, IigP1, Serping1, Srgn, Psmb8, Fkbp5, Ggta1, Gbp2, Amigo2, Fbln5* (LPS group); and *Emp1, Clcf1, Slc10a6, Cd109, Cd14, Ptx4, S100a10, Ptgs2, B3gnt5, Tm4sf1, Sphk1, Tgm1* (ischemic stroke group); and *Gfap, Lcn2, Serpina3n, Aspg1, Cxcl10, Timp1* (pan-reactive markers). We observed increased expression of all pan-reactive markers tested in the *Srf*$^{GFAP-ER}$CKO mouse brain compared to control mice (*Figure 2C*). However, there were no discernible differences in the expression of genes that are upregulated in response to LPS versus ischemic stroke in the *Srf*$^{GFAP-ER}$CKO mice. We found 6 of 12 genes in the LPS group and 9 of 12 genes in the stroke group were upregulated in the *Srf*$^{GFAP-}$ $^{ER}$CKO mice (*Figure 2C*). Together, this indicated that *Srf*-deficient astrocytes exhibit a heterogenous reactive-like phenotype.

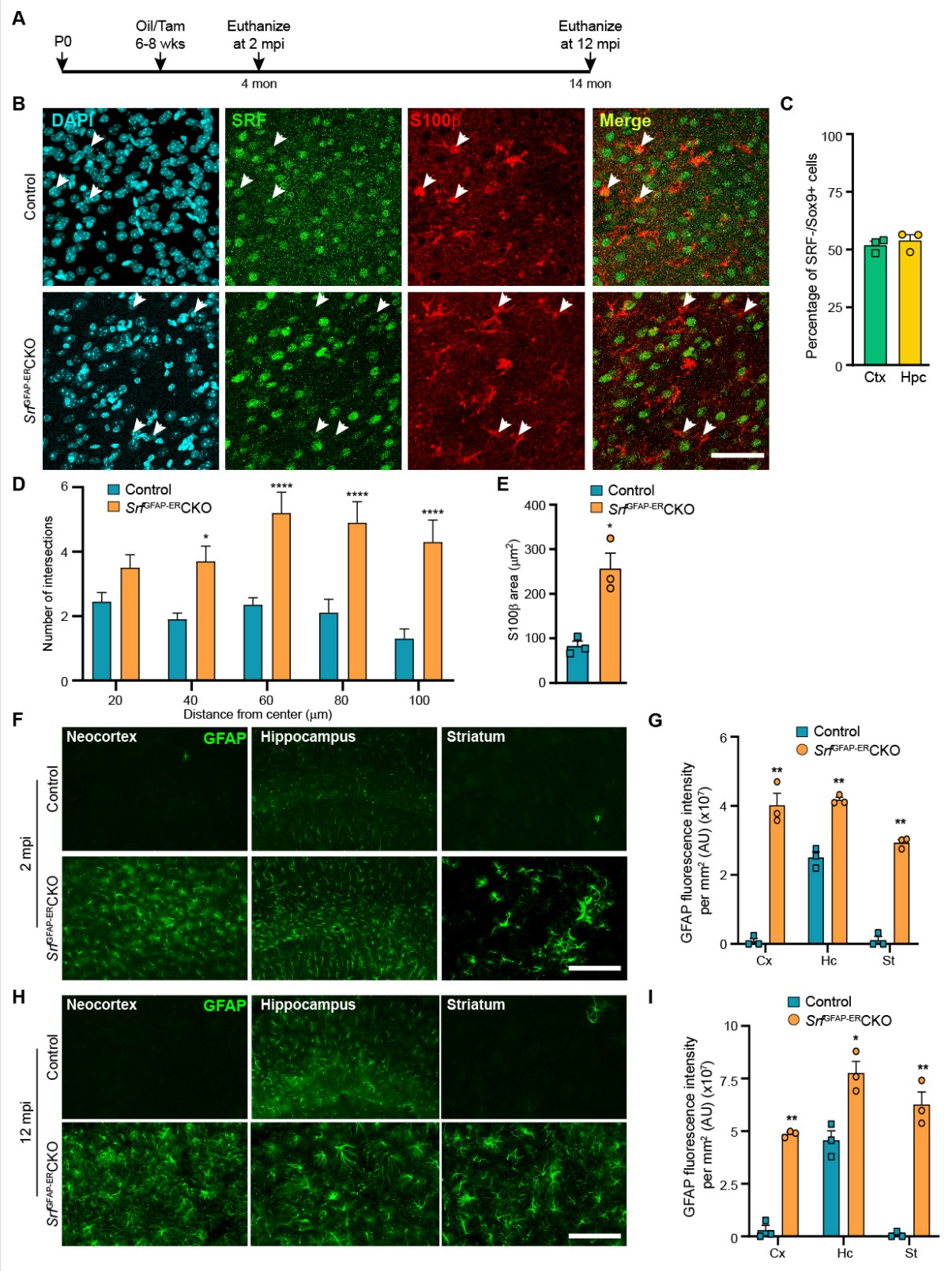

**Figure 1.** *Srf* deletion in post-natal astrocytes results in hypertrophic GFAP +astrocytes. (**A**) Schematic of timeline of tamoxifen injection and analysis of astrocyte phenotype. (**B**) Coronal sections showing immunostaining for SRF and S100β in control and *Srf*GFAP-ERCKO mice at 2 mpi. Astrocytes in control mice showed staining for both SRF and S100β while several astrocytes in the mutant mice did not show any SRF staining (white arrowheads). (**C**) Quantification of extent of SRF loss in astrocytes. (**D**) Sholl analysis of S100β expressing astrocytes from (**B**) shows hypertrophic morphology of SRF

*Figure 1 continued on next page*

*Figure 1 continued*

ablated astrocytes. n=3 mice. (**E**) Quantification of S100β-immunoreactive surface area. n=3 mice. (**F**) Coronal sections showing immunostaining for GFAP in control and *Srf*GFAP-ERCKO mutant mice at 2 mpi. (**G**) Quantification of relative GFAP fluorescence intensity in F. n=3 mice. (**H**) Coronal sections showing immunostaining for GFAP in control and *Srf*GFAP-ERCKO mutant mice at 12 mpi. (**I**) Quantification of GFAP fluorescence intensity in H. n=3 mice. Data are represented as mean ± SEM. * p<0.05, ** p<0.01, **** p<0.0001, ns, not significant. Unpaired t-test. Scale bar, 50 µm (**B**), 100 µm (**F, H**).

The online version of this article includes the following figure supplement(s) for figure 1:

**Figure supplement 1.** Cre recombinase expression in hGFAP-Cre-ERT transgenic mouse line.

**Figure supplement 2.** DAPI staining of brain sections from control and *Srf*GFAP-ERCKO mice at 2 mpi (**A**) and 12 mpi (**B**) for the GFAP staining data shown in *Figure 2*.

## *Srf*-deficient astrocytes do not induce cell death in the brain

Since the *Srf*-deficient astrocytes showed expression of several marker genes associated with astrocyte reactivity, we next sought to determine any deleterious effects caused by *Srf*-deficient astrocytes. Brains sections from *Srf*GFAP-ERCKO mice and control littermates at 2 mpi and 12 mpi (4 months and 14 months of age, respectively) were stained using FluoroJade-C or TUNEL. We did not observe any discernible TUNEL- or FluoroJade-C-positive cells in the brains of *Srf*GFAP-ERCKO mice compared to control littermates (*Figure 2—figure supplement 1*). To confirm the absence of cell loss, immunostaining for the neuronal marker, NeuN, showed no difference in the number of NeuN-positive cells in the neocortex, striatum, and hippocampus of *Srf*GFAP-ERCKO mice at 12 mpi (*Figure 3AB*). Similarly, immunostaining for the oligodendrocyte lineage marker, Olig2, revealed no difference in the number of Olig2+ cells in *Srf* mutant mice at 2 mpi relative to controls (*Figure 3—figure supplement 1*). We next sought to determine whether this prolonged gliosis affected myelination in the brains of *Srf*GFAP-ERCKO mice. Black gold-II staining showed no discernible differences in myelin in the neocortex and hippocampus of *Srf*GFAP-ERCKO mice compared to control mice, even at 12 mpi (*Figure 3—figure supplement 1*). Consistent with the lack of cell death, no gross morphological abnormalities were found in the brains of *Srf* cKO mice at 12 mpi (14–15 months of age; *Figure 3A*). Furthermore, the *Srf*GFAP-ERCKO mice did not exhibit any difference in body weight at 12 mpi compared to control littermates (*Figure 3—figure supplement 1*). Taken together, these data demonstrate that SRF deletion in astrocytes does not cause neuronal cell death or affect myelination.

Astrocytes play important roles in the maintenance of the blood-brain barrier (BBB) and abnormalities in their morphology or functions are associated with loss of BBB integrity (*Abbott et al., 2006*; *Chapouly et al., 2015*). Given the hypertrophic morphology and altered gene expression of astrocytes in the *Srf* cKO mice, we examined whether *Srf*-deficient astrocytes were compromised in their ability to support BBB integrity. Transcardial injection of a small 10 kDa dextran fluorescein tracer revealed no tracer in the brain parenchyma of *Srf*GFAP-ERCKO mice at 17–18 months of age (15 mpi) compared to control mice (*Figure 3—figure supplement 2*), suggesting no effect on BBB integrity.

Microglia rapidly respond to changes in the environment and there exists an active crosstalk between astrocytes and microglia (*Jha et al., 2019*; *Matejuk and Ransohoff, 2020*). Given the morphological and molecular changes exhibited by *Srf*-deficient astrocytes, we next examined the status of microglia in the *Srf*GFAP-ERCKO mice. In the healthy brain, Iba1 immunostaining shows basal expression in the microglial cell body and in highly ramified processes, indicative of a resting state. We observed a similar pattern of immunostaining, morphology, and fluorescence intensity in *Srf* cKO mouse brains at 2 mpi (*Figure 3—figure supplement 3*). Cell counts further revealed similar numbers of Iba1-positive microglia in the neocortex and hippocampus of *Srf* cKO mice at 2 mpi relative to control littermates (*Figure 3—figure supplement 3*). However, Iba1 immunostaining at 12 mpi revealed an increase in fluorescence intensity accompanied with amoeboid-like shape in *Srf* cKO mice, compared to control mice (*Figure 3—figure supplement 3*). Cell counts at 12 mpi showed an increase in Ibal-positive cells per unit area in the neocortex, hippocampus, and striatum of *Srf* cKO mice relative to control mice (*Figure 3—figure supplement 3*). Moreover, neither astrocytes nor microglia were actively proliferating (Ki67 immunoreactivity; *Figure 2—figure supplement 1*) and the increased microglial cell numbers in the *Srf* cKO mice at 12 mpi likely occurred at an earlier time point.

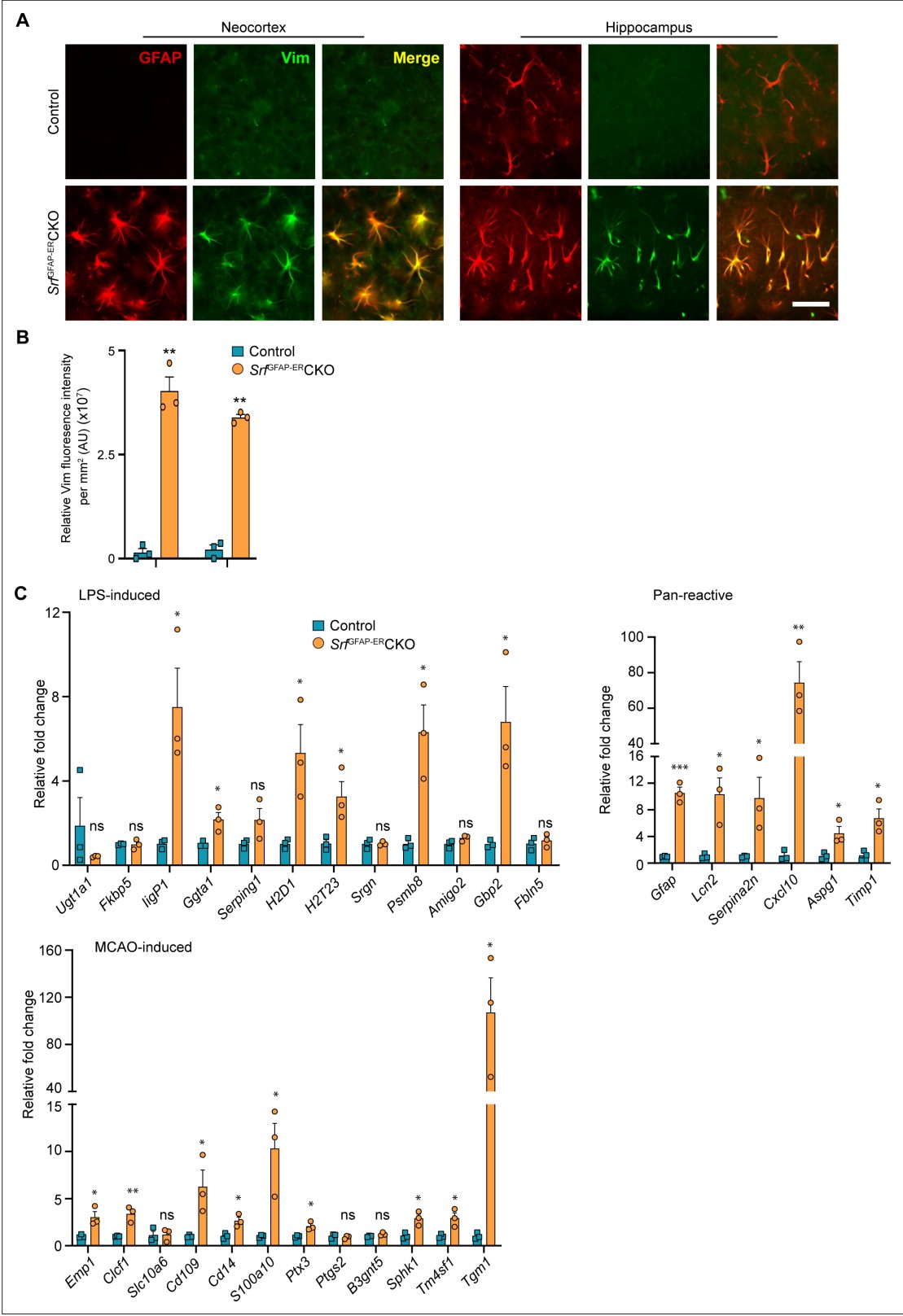

**Figure 2.** *Srf*-deficient astrocytes express reactive astrocyte markers. (**A**) Coronal sections showing GFAP and Vimentin (Vim) immunostaining in control and *Srf*GFAP-ERCKO mice at 2 mpi. Astrocytes in the neocortex and hippocampus were positive for both GFAP and Vimentin. The control astrocytes exhibited weaker staining for GFAP only in the hippocampus and not for Vimentin. (**B**) Quantification of Vim fluorescence intensity in the astrocytes in the neocortex and hippocampus shown in (**A**). n=3 mice. (**C**) Quantitative PCR analysis of genes induced by inflammatory LPS stimulation, middle

*Figure 2 continued on next page*

*Figure 2 continued*

cerebral artery occlusion (MCAO) and markers of pan-reactive astrocytes in control and *Srf* cKO mice. n=3 mice. ** p<0.01. Unpaired t-test. Data are represented as mean ± SEM. Cx, neocortex; Hc, hippocampus. AU, arbitrary units. Scale bar, 25 μm.

The online version of this article includes the following figure supplement(s) for figure 2:

**Figure supplement 1.** Absence of proliferation and cell death in *Srf*<sup>GFAP-ER</sup>CKO mice.

---

## *Srf*<sup>GFAP-ER</sup>CKO mice do not exhibit deficits in synaptic plasticity and behavior

Astrocytes play major roles in synapse formation, maintenance, and elimination (***Augusto-Oliveira et al., 2020***; ***Baldwin and Eroglu, 2017***; ***Chung et al., 2015***). We therefore asked whether *Srf*-deficient astrocytes are compromised in their ability to produce prosynaptogenic factors or affect synapse maintenance. First, we assessed mRNA expression of several astrocyte-secreted synaptogenic factors (***Allen and Eroglu, 2017***). Quantitative real-time PCR analyses showed no change in the expression levels of *Hevin, Glycipan 4/6, Thbs1,* and *Thbs2*, suggesting that the *Srf*-deficient astrocytes are not compromised in their ability to make pro-synaptogenic factors (***Figure 3C***). Second, we investigated whether synapse numbers were altered in the *Srf*<sup>GFAP-ER</sup>CKO mice. Brain slices from 3 mpi and 15 mpi control and *Srf*<sup>GFAP-ER</sup>CKO mice were co-labeled with Piccolo (presynaptic marker) and GluA1 (post-synaptic marker) antibodies, and the number of structural synapses quantified based on overlapping puncta staining (***Harris and Weinberg, 2012***). Similarly, there was no difference in the number of structural synapses in *Srf*<sup>GFAP-ER</sup>CKO mice relative to control littermates (***Figure 3D, E***), arguing that synapses are not affected by *Srf*-deficient astrocytes. Third, we asked whether synaptic transmission or synaptic plasticity are affected in *Srf*<sup>GFAP-ER</sup>CKO mice. Basal synaptic transmission, paired-pulse ratio, and long-term potentiation (LTP) were measured in hippocampal slices (Schaffer-collateral pathway) obtained from *Srf*<sup>GFAP-ER</sup>CKO and control littermates at 3 mpi and 15 mpi (***Booth et al., 2014***; ***Zaman et al., 2000***). There were no differences in paired-pulse ratio, post-synaptic response to stimulus intensity and summated action potential between *Srf*<sup>GFAP-ER</sup>CKO mice and control littermates (***Figure 4—figure supplement 1***). We did not observe any significant differences in either basal synaptic transmission or LTP in the knockout mice compared to their control littermates (3 mpi: synaptic transmission, p=0.3698; LTP, p-value = 0.5306; 15 mpi: synaptic transmission, p=0.1411; LTP, p-value = 0. 07334) (***Figure 4A–D*** and ***Figure 4—figure supplement 1***). Taken together, these findings indicate that *Srf*-deficient astrocytes do not affect synaptic transmission or synaptic plasticity.

We next examined the effect of SRF deficiency on hippocampus-dependent spatial memory. To control for the confounding effects of possible changes in locomotor behavior, we performed an open field test and found no significant differences in the total distance traveled between *Srf*<sup>GFAP-ER</sup>CKO mice and the control littermates (control, 24.83±1.85 m; *Srf*<sup>GFAP-ER</sup>CKO, 26.05±3.10 m; p=0.74, unpaired t-test; ***Figure 4E***). Next, we used the contextual fear-conditioning paradigm to assess spatial memory in *Srf*<sup>GFAP-ER</sup>CKO mice and control littermates at 9 mpi. 24 hr following fear conditioning in a shock context (context A), animals were tested for their freezing response in the same context, followed by context B which served as a control environment which wasn't paired with the foot shock. While we found a statistically significant difference in the percent time spent freezing between context A and context B across genotypes (F (1, 10)=26.42, p=0.0004), we did not find any difference between the genotypes in their freezing responses (Context A: control, 32.44±8.20; *Srf*<sup>GFAP-ER</sup>CKO, 36.53±10.99, p=0.91, Context B: control, 12.51±4.37; *Srf*<sup>GFAP-ER</sup>CKO, 13.87±5.91, p=0.99, two-way ANOVA, Sidak's post hoc test; ***Figure 4F***). The mice were tested again in the same contexts (context B and context A) for remote recall after 1 month from fear conditioning. Again, while both genotypes could differentiate between the 2 contexts (F (1, 10)=15.73, p=0.0027), we found no difference in the percent time spent freezing between *Srf*<sup>GFAP-ER</sup>CKO mice and control littermates in either of the contexts (Context B: control, 26.41±8.55; *Srf*<sup>GFAP-ER</sup>CKO, 31.64±13.42, p=0.95; Context A: control, 45.40±11.86, *Srf*<sup>GFAP-ER</sup>CKO, 54.31±17.08, p=0.86, two-way ANOVA, Sidak's post hoc test) (***Figure 4F***).

Finally, we assessed spatial memory in the Barnes maze test. Animals were trained to locate an escape box on a Barnes maze until the learning curve flattened. Mice were tested in a probe test in the absence of an escape box, 2 hr after the last trial, and the latency to locate the target hole and the time spent in the target quadrant were calculated. We did not find a significant difference in the latency to target hole location (control, 35.95±8.08, *Srf*<sup>GFAP-ER</sup>CKO: 26.22±5.90, p=0.35, unpaired

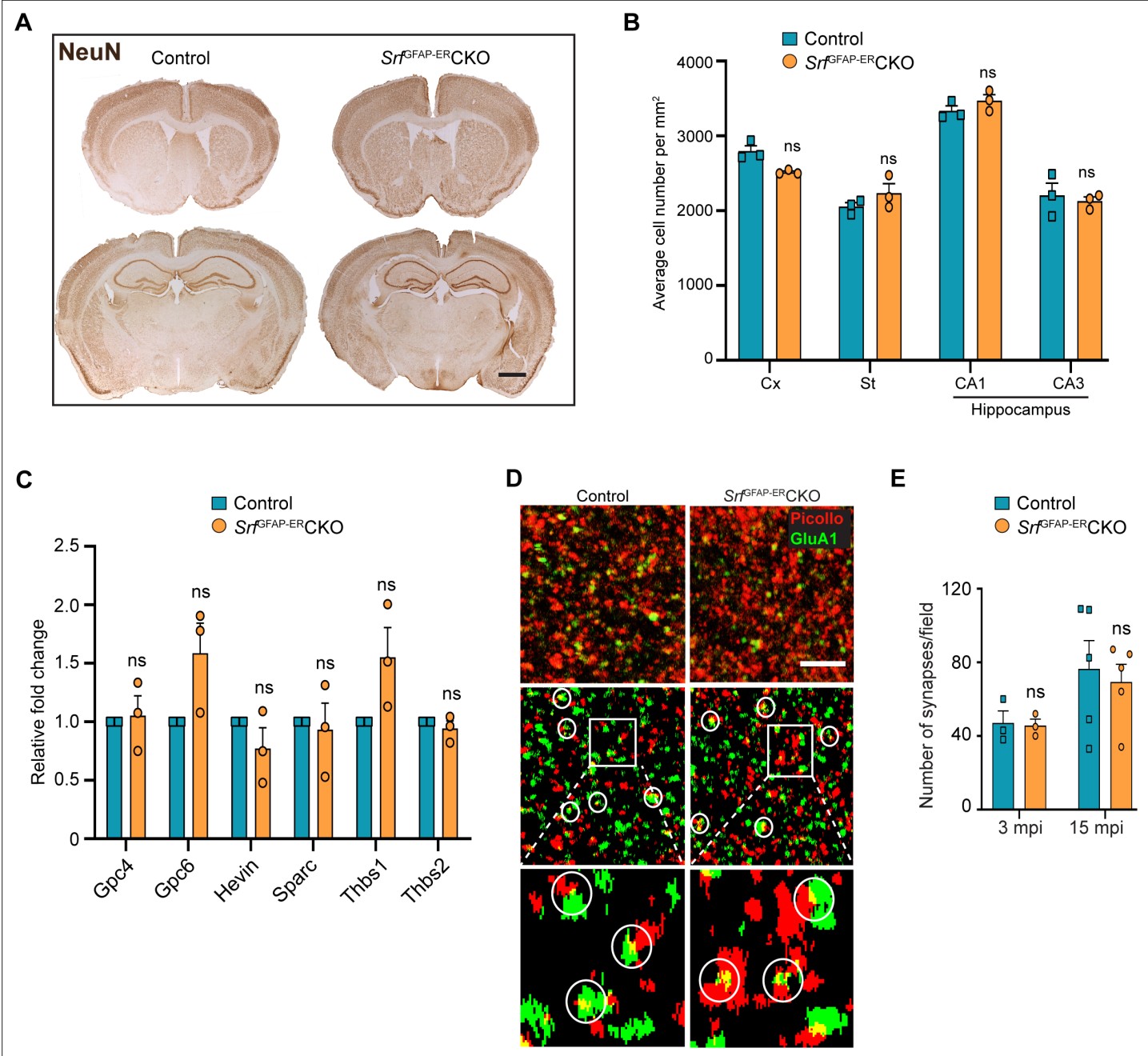

**Figure 3.** *Srf*-deficient astrocytes do not affect neuronal survival and synapse numbers. (**A**) Coronal sections showing NeuN immunostaining in control and *Srf* mutant mice at 12 mpi. (**B**) Quantification of NeuN+ cell numbers in the neocortex and striatum, and DAPI+ cell numbers in the CA1 and CA3 regions of the hippocampus. (**C**) Quantitative PCR of astrocyte-secreted synaptogenic factors shows no difference in their expression in the mutant mice when compared to control mice. n=3 mice. (**D**) Representative images of neocortical sections immunostained for the presynaptic marker, piccolo (red) and postsynaptic marker, GluA1 (green). Co-localization of staining (yellow puncta) was counted as a synapse. Scale bar, 10 μm. (**E**) Quantification of the number of synapses in the neocortex in control and *Srf*^GFAP-ER^CKO mutant mice at 3 and 15 mpi. n=3 mice. Unpaired t-test. Data are represented as mean ± SEM. ns, not significant.

The online version of this article includes the following figure supplement(s) for figure 3:

**Figure supplement 1.** Oligodendrocyte lineage cells and myelination in *Srf*^GFAP-ER^CKO mice.

**Figure supplement 2.** BBB integrity is not compromised in *Srf*^GFAP-ER^CKO mutant mice.

**Figure supplement 3.** Increase in Iba1 intensity and cell numbers in older but not younger *Srf* mutant mice.

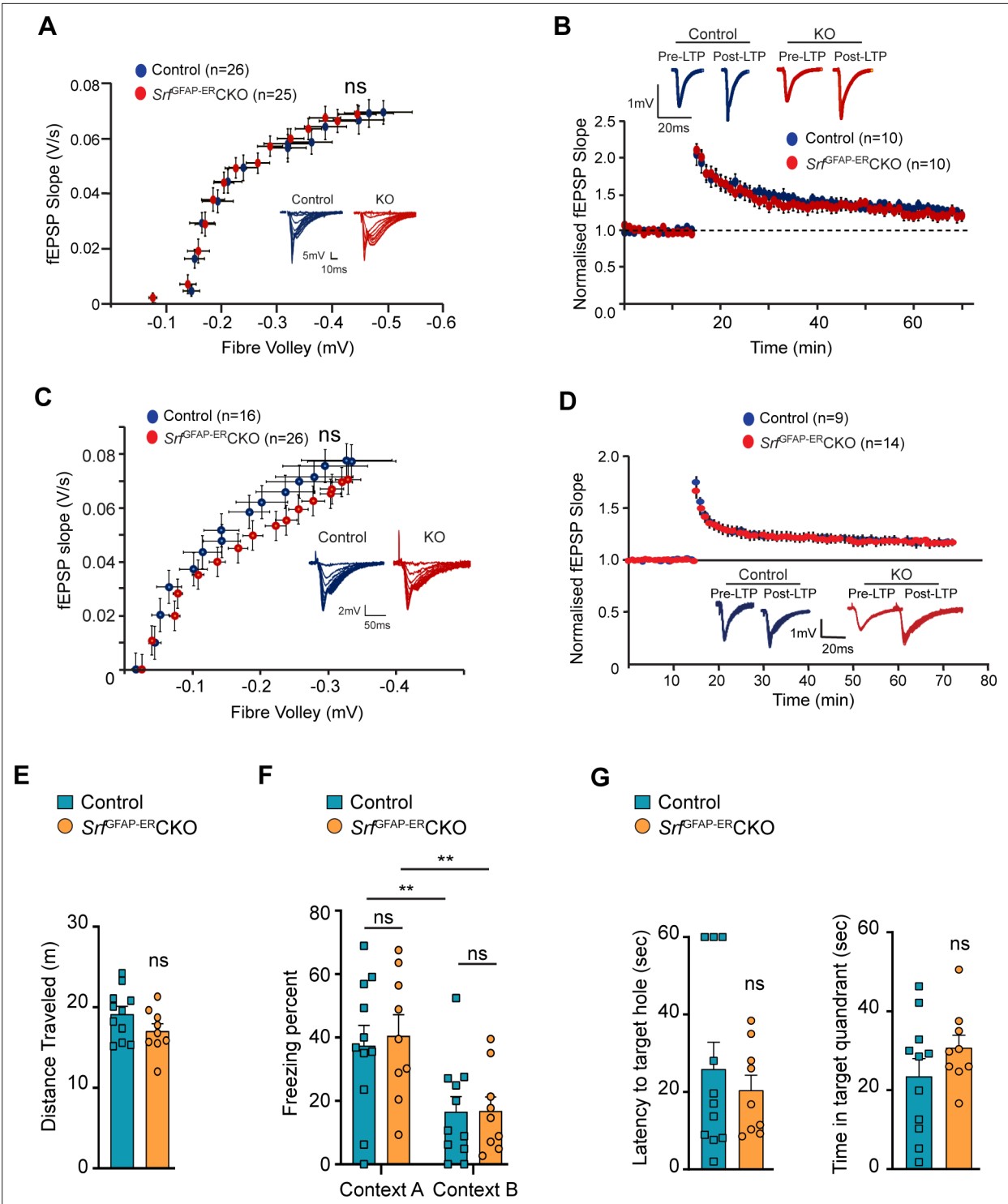

**Figure 4.** SRF-deficient astrocytes support normal synaptic plasticity, learning and memory. (**A–D**) Electrophysiological measurements showed no significance between control and mutant mice. (**A**) Basal synaptic transmission, (**B**) LTP in the hippocampus at 3 mpi, (**C**) Basal synaptic transmission, (**D**) LTP in the hippocampus at 15 mpi. Example traces are those recorded for 1–2 min around the time point indicated by I and II in the graph. The number of slices recorded from are indicated in parentheses. (**E–G**) Behavioral analyses. (**E**) Open-field test, (**F**) contextual fear conditioning at recent and remote time points in context A (shock context) and context B (no-shock context), and (**G**) Barnes maze test at 9 mpi in control and mutant mice. n=7 (control), n=5 (mutant) mice. Data are represented as mean ± SEM. ns, not significant. Unpaired t-test; two-way ANOVA, Sidak's post hoc test (**F**).

The online version of this article includes the following figure supplement(s) for figure 4:

**Figure supplement 1.** Normal synaptic functions in *Srf*^GFAP-ER^CKO mice.

t-test) and the time spent in the target quadrant between $Srf^{GFAP-ER}$CKO mice and control littermates (control, 20.96±6.90; $Srf^{GFAP-ER}$CKO, 29.60±5.64, p=0.35, unpaired t-test) (*Figure 4G*). Collectively, these experiments establish that the persistent and widespread presence of *Srf*-deficient astrocytes is not detrimental to normal synaptic plasticity or learning and memory.

## Transcriptomic profile of *Srf*-deficient astrocytes

To gain a better understanding of the molecular nature of *Srf*-deficient astrocytes, we performed RNA sequencing (RNA-seq) analysis of astrocytes isolated from the forebrain of 5-week-old $Srf^{GFAP}$CKO (*Jain et al., 2021*) and 4 mpi (6-month-old) $Srf^{GFAP-ER}$CKO mice and their respective control littermates. Genes with fold change of >1.5 and adjusted p-value <0.05 were considered significant. The expression profile of differentially expressed genes (DEGs) across the samples is presented in volcanoplots (*Figure 5A*; *Figure 5—source data 1 and 2*). There were 827 significantly expressed genes (726 upregulated and 101 downregulated) in the $Srf^{GFAP-ER}$CKO data set and 903 sigificantly expressed genes (621 upregulated and 282 downregulated) in the $Srf^{GFAP}$CKO data set (*Figure 5—source data 1 and 2*). There were 315 common up-regulated genes in the $Srf^{GFAP}$CKO and $Srf^{GFAP-ER}$CKO astrocytes. Enrichment analysis of Gene Ontology terms for Biological Process (GO BP) indicated that SRF deficiency results in enrichment of pathways related to immune response and innate immunity (*Cd86, H2-T23, Cd84, Lst1, Ifit3, Trim30a, Trim30b*), inflammatory response (*Nlrp1b, Ccl12, Cxcl13, Ccl5, Ccl2, Cd14*), antiviral defense (*Zbp1, Rsad2, Mx1, Oas1a, Oas1g, Oas2*), response to interferon-beta and regulation of interleukin-1 beta (*Figure 5B, D*; *Figure 5—source data 3*). There was also an enrichment of genes related to microglial cell activation (*Tlr1, C1qa. Itgam, Tlr6, Tlr2*) (*Figure 5B, D*; *Figure 5—source data 3*).

Analysis of downregulated genes revealed pathways related to cell adhesion (*Pcdhgb5, lamc1, Hapln3, Celsr2*), extracellular matrix (*Ecm2, Eln, Dmp1, Fnln1*), lipid metabolism (*Slc27a1, Acaa2, Msmo1, Aacs, FADS2*), Wnt signaling pathway (*Yap1, Fzd1, Fzd2, Wnt7b*), smoothened signaling pathway (*Evc2, Ptch1, Tulp3, Pax6, Gpr37l1, Gli1, Gli3, Gli2*) and Notch signaling pathway (*Sox2, Dll4, Enho, Yap1, Notch1, Hes1, Ascl1, Hes5*) (*Figure 5C*; *Figure 5—source data 3*). Genes involved in glial cell differentiation (*Notch1, Chd2, Erbb2, Ascl1, Klf15*) were also downregulated, consistent with our earlier observations on the role of SRF in regulating glial differentiation (*Lu and Ramanan, 2012*). Under certain conditions, reactive astrocytes have been shown to cause cell death of neurons and oligodendrocytes partly through secretion of neurotoxic factors such as lipocalin 2 (*Lcn2*) and saturated fatty acids (*Bi et al., 2013*; *Guttenplan et al., 2021*). We analyzed our expression data set and found that the expression of Lcn2, C3 and the elongase of very-long fatty acid (Elovl) family of genes (involved in the synthesis of longer-chain saturated fatty acids) either remained unchanged or were downregulated in *Srf*-deficient astrocytes (*Figure 5—source data 3*). Together, these data suggested that *Srf*-deficient astrocytes exhibit a distinct transcriptome, and do not exhibit upregulation of genes shown to be involved in neurotoxicity.

## *Srf*-deficient astrocytes show up-regulation expression of genes involved in neuroprotection

Next, we analyzed the above astrocytic RNA sequencing data for genes that could provide insights into the potential functional nature of these astrocytes. The transmembrane chemokine CXCL16 has been shown to promote physiological neuroprotection following ischemic and excitotoxic insults via astrocytic release of CCL2 and adenosine 3 receptor (Adora3/A3R; *Rosito et al., 2012*; *Rosito et al., 2014*). Expression of *Cxcl16, Ccl2* and *Adora3/A3R* expression was increased in *Srf*-deficient astrocytes (log₂fold change: *Cxcl16*, 4.3-fold; *Ccl2*, 4.1-fold; *Adora3*, 3.3-fold; *Figure 5D*; *Figure 5—figure supplement 1*). Insulin-like growth factor1 (IGF-1) was also increased 3.2-fold, which has been shown to provide neuroprotection in models of kainic-acid induced excitotoxicity (*Chen et al., 2019*). We also found an upregulation of genes involved in oxidative defense system, namely glutathione peroxidase-1 (GPx1) (twofold) and heme oxygenase 1 (*Hmox1/HO-1*) (1.8-fold) (*Figure 5—figure supplement 1*). Enrichment analysis of Gene Ontology terms for Biological Process (GO BP) revealed that *Srf*-deficient astrocytes showed enrichment of pathways related to cellular response to beta-amyloid and beta-amyloid clearance (*Figure 5—figure supplement 1*). Among the cellular response to beta-amyloid genes, *NAMPT* (2.9-fold), *Casp4* (4.6-fold), *Trem2* (5.5-fold), *Igf1* (3.2-fold), and *Adrb2* (2.4-fold) were upregulated. In addition, genes that promote Aβ uptake and clearance were also increased.

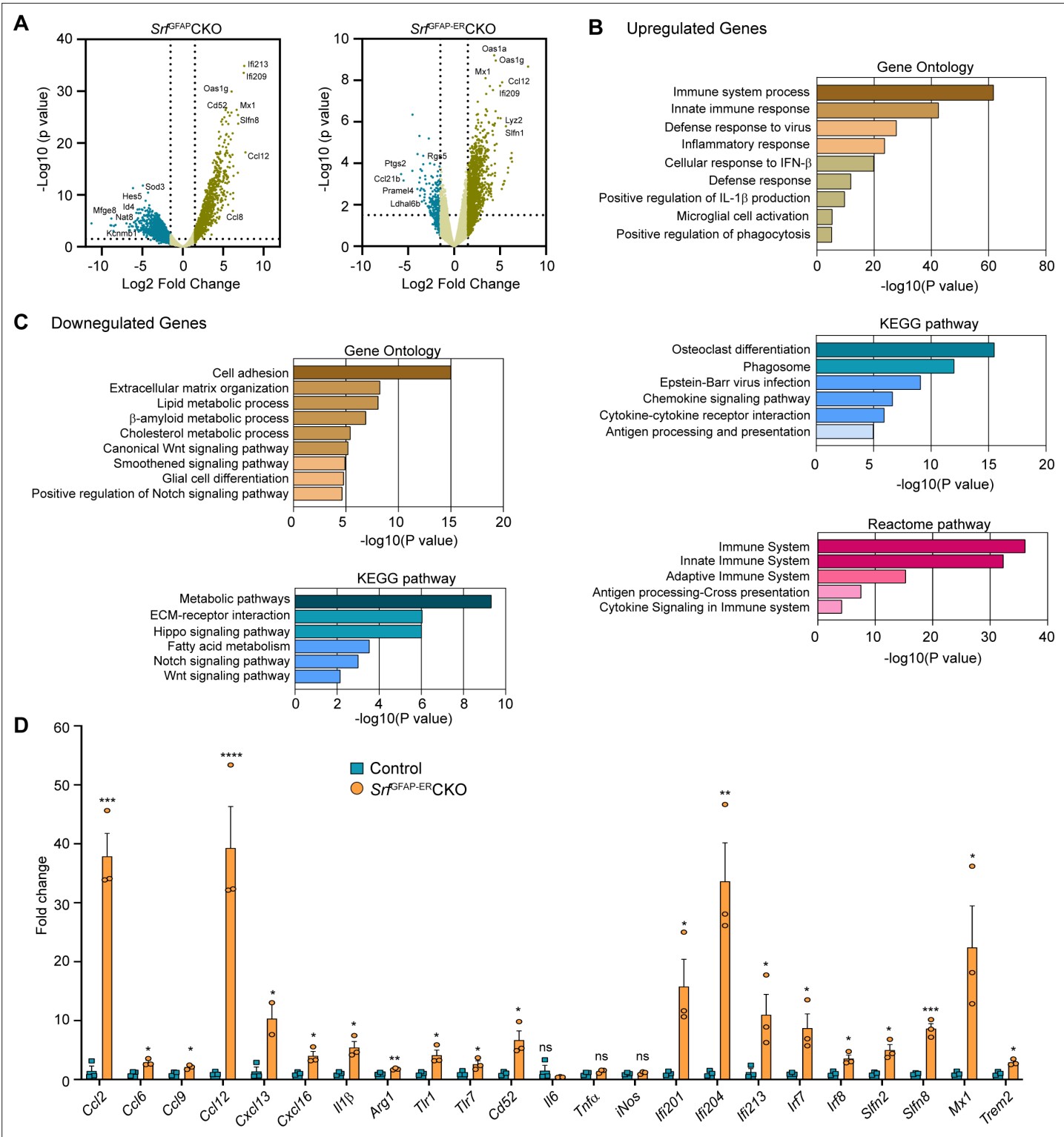

**Figure 5.** Transcriptome profile of *Srf*-deficient astrocytes. (**A**) Volcano plots of differentially expressed genes in control and *Srf*-deficient astrocytes from RNA sequencing. (**B**) Gene Ontology, KEGG pathway and Reactome pathway analysis of upregulated genes in *Srf*-deficient astrocytes. (**C**) Gene Ontology and KEGG pathway analysis of downregulated genes in *Srf*-deficient astrocytes. (**D**) Quantitative PCR analysis of interferon, inflammatory and defense response genes in reactive astrocytes identified in Gene Ontology analysis. n=3 mice. Data are represented as mean ± SEM. * p<0.05, ** p<0.01, *** p<0.005, ****p<0.001, ns, not significant. Unpaired t-test.

The online version of this article includes the following source data and figure supplement(s) for figure 5:

*Figure 5 continued on next page*

*Figure 5 continued*

**Source data 1.** Top 25 genes up- and downregulated in SrfGFAPCKO astrocytes.

**Source data 2.** Top 25 genes up- and downregulated in SrfGFAP-ERCKO astrocytes.

**Source data 3.** KEGG pathway analysis of Srf-deficient astrocytes.

**Figure supplement 1.** Differential expression of genes involved in neuroprotection in *Srf* mutant astrocytes.

These include lipoprotein lipase (*Lpl*) (4-fold), *Hmox1/HO-1* (1.8-fold), the member of the superfamily of ATP-binding cassette (ABC) transporters *Abca1* (1.7-fold) and beta-2-adrenergic receptor (*Adrb2*) (2.4-fold) and IGF-1 (*Figure 5—figure supplement 1*). Also, there is a downregulation of genes such as *Bace2* (–3.5-fold) and *Srebf2* (–2.1-fold), which are involved in β-amyloid metabolic process (*Figure 5—figure supplement 1*).

### *Srf*^GFAP-ER^CKO mice exhibit reduced excitotoxic cell death

We finally sought to determine whether *Srf*-deficient astrocytes could provide neuroprotection in response to neuronal insult or in the setting of disease. Neuronal excitotoxicity is a major cause of cell death in CNS injuries and neurodegenerative diseases (*Binvignat and Olloquequi, 2020*). Kainic acid (KA) causes neurotoxic cell death of mainly the hippocampal CA1 and CA3 pyramidal neurons (*Pollard et al., 1994*). We asked whether SRF-deficient astrocytes could protect neurons from KA-induced excitotoxicity. We administered KA via intracerebroventricular injection into the lateral ventricles of control and mutant mice at 4 mpi (*Figure 6A*) and assessed cell death using TUNEL and FluoroJade-C staining at 7 days post-KA injection. While control mice showed numerous TUNEL +and FluoroJade-C + cells in the CA1, CA3 and dentate gyrus, indicative of extensive neuronal cell death (*Figure 6B and C* and *Figure 6—figure supplement 1*), little to no cell death was seen in the hippocampus of *Srf*^GFAP-ER^CKO mice (*Figure 6B, C* and *Figure 6—figure supplement 1*).

### *Srf*-deficient astrocytes protect dopaminergic neurons in a model of Parkinson's disease

Parkinson's disease (PD) is caused by death of midbrain dopaminergic (DA) neurons in the substantia nigra pars compacta region (SNpc). We employed the 6-hydroxydopamine (6-OHDA) model of PD to investigate whether *Srf*-deficient reactive astrocytes can protect DA neurons from cell death (*Ungerstedt, 1968*). We first confirmed the presence of reactive astrocytes in the substantia nigra region in the *Srf* cKO mice before administration of 6-OHDA (*Figure 6—figure supplement 2*). Next, unilateral stereotaxic injection of 6-OHDA was administered into the SNpc of control and mutant mice at 9 mpi and cell death was analyzed at 10 days post-injection (*Figure 6D*). Control mice showed significant loss of tyrosine hydroxylase (TH)-positive DA neurons in the ipsilateral side compared to uninjected contralateral size (*Figure 6E and F* and *Figure 6—figure supplement 2*). In striking contrast, there was little to no DA neuron and fiber loss in the ipsilateral side of *Srf*^GFAP-ER^CKO mice compared to the ipsilateral side of control and to the contralateral side of knockout mice (*Figure 6E, F* and *Figure 6—figure supplement 2*).

### *Srf*^GFAP-ER^CKO/AD transgenic mice exhibit reduced amyloid plaque burden

We next investigated the role of reactive astrogliosis in the APP~Swe~/PSen1~dE9~ (APP/PS1) model of Alzheimer's disease (AD). A morphological hallmark of AD is the deposition of β-amyloid (Aβ) plaques, which is associated with neurodegeneration and cognitive decline. We intercrossed *Srf*^GFAP-ER^CKO mice and APP/PS1 transgenic mice to generate *Srf*^GFAP-ER^CKO; AD^Tg+/-^ triple transgenic mice. *Srf*^GFAP-ER^CKO; AD^Tg+/-^ mice and control mice received tamoxifen injections at 6–8 weeks of age, and all groups of mice were analyzed at 9–12 months of age for Aβ plaques (*Figure 6G*). Immunostaining using the MOAB-2 antibody, which detects only amyloid-β peptide, but not APP (*Youmans et al., 2012*), showed numerous plaques in the neocortex and hippocampus of APP/PS1 control mice (*Srf*^f/f^; AD^Tg+/-^) (*Figure 6H, I*). In contrast, *Srf*^GFAP-ER^CKO; AD^Tg+/-^ mice had 60% fewer plaques compared to APP/PS1 mice (*Figure 6H, I*). In *Srf*^GFAP-ER^CKO; AD^Tg+/-^ mice, reactive astrocytes were present for 5–8 months (until the end of the experiment) starting at 4 months of age. We then sought to determine whether *Srf*-deficient astrocytes, if present for a longer duration, can cause further reduction in the number of

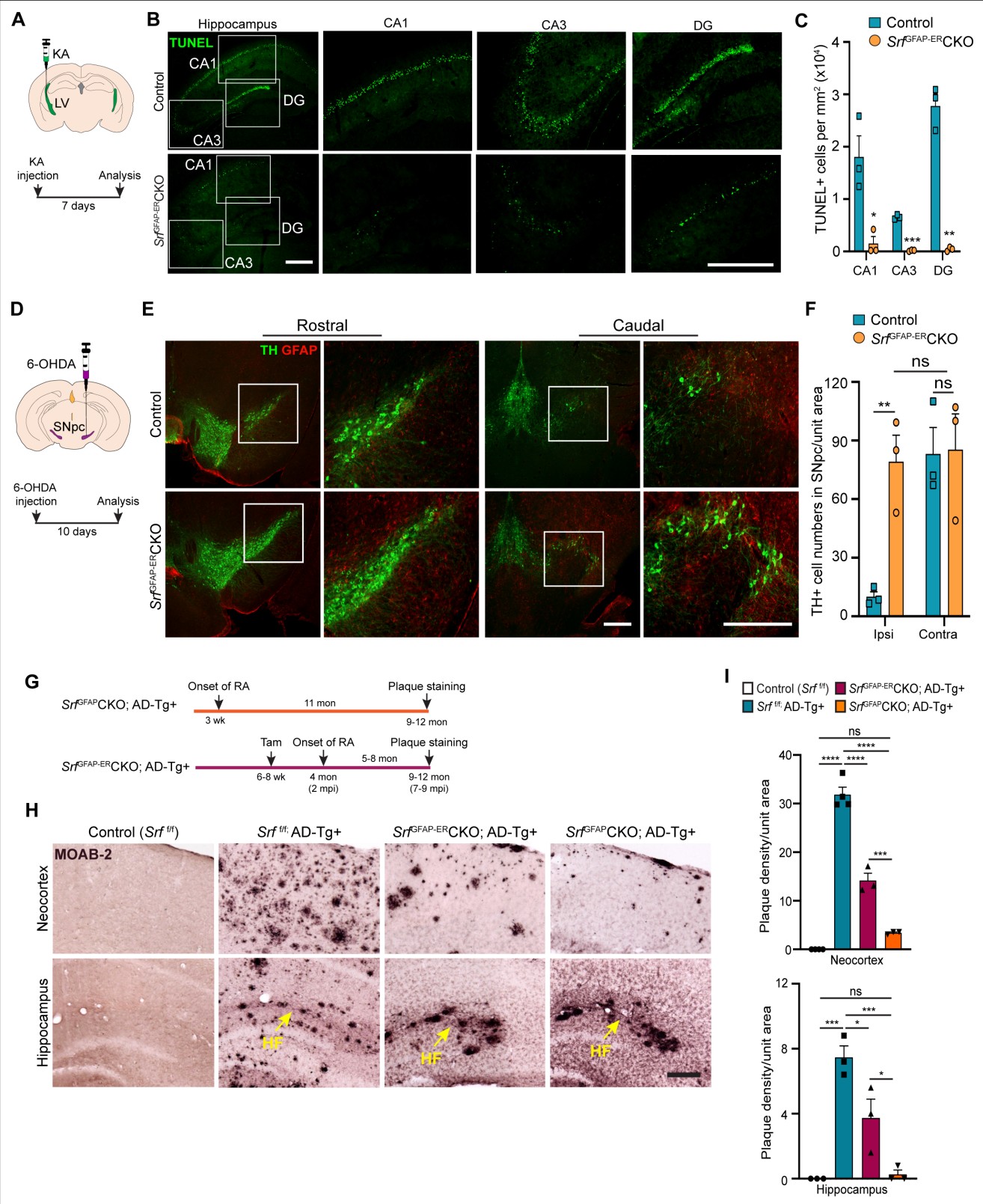

**Figure 6.** *Srf*-deficient reactive astrocytes confer neuroprotection in the brain. (**A**) Schematic of stereotaxic intracerebroventricular (ICV) injection of kainic acid (KA) (top) and timeline of experiment (bottom) in 4 mpi mice. (**B**) Coronal brain sections showing TUNEL (terminal deoxynucleotidyl transferase dUTP nick end labeling) staining at 7 days post-kainic acid injection shows cell death in hippocampal regions CA1, CA3 and DG. Scale bar, 200 µm. (**C**) Quantification of TUNEL+ cells in the hippocampal regions in B. n=4 mice. (**D**) Schematic of stereotaxic injection 6-hydroxydopamine (6-

*Figure 6 continued on next page*

*Figure 6 continued*

OHDA) in the substantia nigra *pars compacta* (SNpc) (top) and timeline of experiment (bottom) in control and mutant mice at 9 mpi. (**E**) Immunostaining of coronal sections showing tyrosine hydroxylase (TH⁺) dopamine neurons and GFAP⁺ astrocytes in the rostral and caudal regions of substantia nigra at 10 days following 6-OHDA injection. Magnified image of the white boxed region is shown on the right. Scale bar, 200 µm. (**F**) Quantification of TH⁺ neurons in the ipsilateral (ipsi) and contralateral (contra) side of (**E**). n=3 mice. (**G**) Schematic showing the timeline of reactive astrocyte (RA) induction and amyloid plaque staining in AD mice. (**H**) Coronal sections showing MOAB2-immunostained β-amyloid plaques. Scale bar, 200 µm. (**I**) Quantification of plaque density from (**H**). RA, reactive astrocytes, LV, lateral ventricular, SNpc, Substantia nigra pars compacta, SR, Stratum radiatum. Yellow arrows show hippocampal fissure, HF. n=3 mice. Data are represented as mean ± SEM. * p<0.05, ** p<0.01, *** p<0.005, **** p<0.001, ns, not significant. Unpaired t-test (**C, F**), One-way ANOVA with Tukey post-hoc test (**I**).

The online version of this article includes the following figure supplement(s) for figure 6:

**Figure supplement 1.** Neuroprotection from kainic acid-induced excitotoxicity.

**Figure supplement 2.** Neuroprotection in Parkinson's disease mouse model.

plaques. To address this, we crossed the *Srf*ᴳᶠᴬᴾCKO mice with APP/PS1 mice to generate *Srf*ᴳᶠᴬᴾCKO; ADᵀᵍ⁺/⁻ transgenic mice. In *Srf*ᴳᶠᴬᴾCKO mice, SRF deletion occurs embryonically (around E16.5), and reactive-like astrocytes are seen starting at 3 weeks of life and persist throughout adulthood (*Jain et al., 2021*). Surprisingly, *Srf*ᴳᶠᴬᴾCKO; ADᵀᵍ⁺/⁻ mice greatly reduced number of plaques in the neocortex and hippocampus (*Figure 6H, I*), demonstrating that *Srf*-deficient astrocytes aid in efficient plaque reduction and that longer presence of these mutant astrocytes promoted greater reduction in amyloid plaques.

## Discussion

Astrocyte reactivity is a critical cellular response that plays a pivotal role in aiding neuronal recovery and restoring normal functions following injury and in neurodegenerative diseases (*Pekny and Pekna, 2014*; *Sofroniew, 2005*). Whether reactive astrocytes generated in response to tissue damage and in the setting of disease are beneficial or detrimental for normal brain health is context-dependent (*Moulson et al., 2021*; *Pekny and Pekna, 2014*; *Sofroniew, 2020*). Mounting evidence has shown that reactive astrocytes often become detrimental to normal neuronal functions and can act as active drivers of neurodegeneration (*Huang et al., 2022*; *Phatnani and Maniatis, 2015*). Therefore, it is critical to identify factors that can modulate astrocyte reactive states to achieve beneficial outcomes and provide therapeutic targets to treat neuronal injuries and neurodegenerative diseases. In this study, we demonstrate that conditional SRF ablation in adult astrocytes causes widespread reactive-like astrocytes in the brain. We show that SRF-deficient astrocytes do not affect neuronal and oligo-dendrocyte survival, BBB integrity, synapse numbers, synaptic plasticity, or learning and memory. Strikingly, these *Srf*-deficient reactive astrocytes protect hippocampal neurons from excitotoxicity and nigral dopaminergic neurons from 6-OHDA-induced cell death, as well as reduce amyloid plaque burden in a mouse model of AD.

SRF is a ubiquitous transcription factor that plays several critical functions in the nervous system (*Tabuchi and Ihara, 2022*). SRF ablation in neurons caused deficits in cell migration, neuronal structure and connectivity, synaptic plasticity and learning and memory (*Johnson et al., 2011*; *Knöll and Nordheim, 2009*; *Lu and Ramanan, 2011*). SRF deletion in developing neural stem cells specifically affected glial differentiation (*Lu and Ramanan, 2012*). Our observations show that SRF deletion in perinatal or adult astrocytes causes astrocyte reactivity-like state in the brain suggesting that SRF plays an important non-cell autonomous role in maintaining astrocytes in a non-reactive state (*Jain et al., 2021*). Several other genes have been reported to regulate astrocyte non-reactive state and in some instances, in a region-restricted manner (*Chen et al., 2018*; *Garcia et al., 2010*; *Kang et al., 2014*; *Lananna et al., 2018*; *Robel et al., 2015*). The reactive-like astrocytes observed in some of these studies have been found to be detrimental to normal brain functions and caused spontaneous seizures or reduced viability (*Chen et al., 2018*; *Robel et al., 2015*). We did not observe any discernible detrimental effects of astrocytic SRF deletion on viability, body weight, general behavior, and neuronal functions.

Reactive astrocytes, depending on the context of injury and neurodegeneration, exhibit a spectrum of phenotypic changes that could alleviate or exacerbate neuropathology (*Moulson et al., 2021*). Reactive astrocytes observed in neuroinflammatory conditions and neurodegenerative disorders

are neurotoxic and have degenerative influences on the brain (*Boisvert et al., 2018*; *Chun et al., 2020*; *Clarke et al., 2018*; *Liddelow et al., 2017*; *Yun et al., 2018*). In fact, inhibition of the formation of these neuroinflammatory reactive astrocytes has been found to be beneficial for neuronal survival following injury and attenuated neurodegeneration in mouse models of Alzheimer's disease, Parkinson disease, and ALS (*Guttenplan et al., 2020a*; *Guttenplan et al., 2020b*; *Mann et al., 2022*; *Park et al., 2021*; *Yun et al., 2018*). On the other hand, reactive astrogliosis following ischemic stroke and traumatic brain injury exhibits neuroprotective functions (*Aswendt et al., 2022*). Our findings show that *Srf*-deficient astrocytes are neuroprotective in the brain. These mutant astrocytes efficiently protected hippocampal neurons from excitotoxicity. The cell populations that are most vulnerable to systemic kainic-acid-induced cell death are the hippocampal CA3 and CA1 pyramidal neurons while the dentate granule neurons are usually spared (*Sperk, 1994*). We found that intracerebroventricular injection of kainic acid caused death of neurons in all the regions including the dentate gyrus in the control mice and this was attenuated in the *Srf* cKO mice. Death of CA3 neurons is due to excitotoxicity while that of CA1 neurons is thought to be due to anoxia/ischemia caused by status epilepticus (*Sperk, 1994*). The decreased cell death in these regions in the *Srf* cKO mice suggests that *Srf*-deficient astrocytes likely protect neurons from both anoxic/ischemic and excitotoxic cell death. 6-OHDA is an oxidizable analog of dopamine that causes death of nigral dopaminergic (DA) neurons via oxidative stress (*Hernandez-Baltazar et al., 2017*). The lack of any discernible cell death of DA neurons following 6-OHDA administration in the *Srf* cKO mice revealed robust neuroprotection from oxidative stress. In addition, *Srf*-deficient astrocytes in the APP/PS1 model of AD resulted in a greater reduction of amyloid plaque burden in the cortex and hippocampus. A recent study has shown that SRF may contribute to gene expression changes in reactive astrocytes in a broad cross-section of CNS disorders (*Burda et al., 2022*). These observations together suggest that SRF is a major mediator of reactive astrocytes that could provide neuroprotection in the brain following excitotoxity and in neurodegeneration.

There are some limitations to the current study. We have shown that the *Srf* cKO mice do not exhibit any gross structural abnormalities or deficits in synaptic plasticity and behavior. However, it is possible that they may exhibit as yet uncharacterized structural or behavioral abnormalities. Astrocyte-restricted β1-integrin deletion results in widespread reactive astrocytes but these mutant mice showed spontaneous seizures (*Robel et al., 2015*). Although there was no obvious occurrence of seizures in the *Srf* cKO mice, electroencephalography recordings may provide information on any neuronal hyperexcitability in these mutant mice. Another limitation is that the molecular mechanisms by which *Srf*-deficient astrocytes provide neuroprotection is not known. The *Srf*-deficient astrocytes upregulate genes involved in regulating oxidative stress and these, including *Cxcl16-Ccl2-Adora3*, along with *Gpx1, Hmox1* could protect neurons from kainate-induced excitotoxicity and following 6-OHDA administration. We also observed a significant reduction in Aβ plaque burden, and this could be due to either deficits in Aβ production or efficient clearance by astrocytes. Several genes involved in Aβ clearance were upregulated in *Srf*-deficient astrocytes suggesting that astrocytes may promote clearance. We observed a greater accumulation of plaques near the cortical pial surface and in the hippocampal fissure region and these could be potential regions through which Aβ is cleared. Microglia can also become reactive in response to injury and neurodegeneration and reactive microglia can exert beneficial or detrimental effects on neuronal repair and recovery (*Muzio et al., 2021*). The molecular or functional nature of the reactive microglia observed in the *Srf* cKO mouse brains is not known. The absence of any structural and behavioral abnormalities in the *Srf*GFAP-ERCKO mice strongly suggests that altered microglial morphology and increased numbers in these mice are unlikely to be detrimental and could potentially contribute to neuroprotection and Aβ clearance.

In summary, using the SRF transcription factor as a key regulator of reactive astrocytes, we demonstrate that the persistent presence of *Srf*-deficient astrocytes is not detrimental to brain cell survival, architecture, synaptic plasticity, learning or memory. Importantly, in the setting of both excitotoxicity and neurodegenerative disease these *Srf*-deficient reactive astrocytes provide significant neuroprotection and thus making astrocytic SRF a potential therapeutic target. In addition, the identification of SRF-dependent genes and pathways and the elucidation of neuroprotective mechanisms may open the door for novel pharmacologic therapies aimed at converting astrocytes to a neuroprotective state and allowing for optimized neuronal recovery following injury or in the setting of neurodegenerative disease.

## Materials and methods

### Animals

*Srf*-floxed mice were previously described (**Ramanan et al., 2005**). These mice were bred with GFAP-Cre (generously provided by Dr. David Guttmann, Washington University School of Medicine, St. Louis, MO) and GFAP-Cre-ERT (generously provided by Dr. Suzanne J. Baker, St. Jude Children's Research Hospital, Memphis, TN) transgenic mice to obtain *Srf*[f/f; GFAP-Cre+/-] (*Srf*[GFAP]CKO) and *Srf*[f/f;GFAP-CreERT+/-] (*Srf*[GFAP-ER]CKO) respectively. *Srf*[f/f] mice served as control in all experiments. Control and mutant mice were housed together, and cage mates were randomly assigned to experimental groups. All the procedures in this study were performed according to the rules and guidelines of the Committee for the Purpose of Control and Supervision of Experimental Animals (CPCSEA), India. The research protocols CAF-Ethics-596–2018 and CAF-Ethics-731–2020 were approved by the Institutional Animal Ethics Committee (IAEC) of the Indian Institute of Science.

### Tamoxifen treatment

Tamoxifen (Alfa Aesar, Cat. No. J63509-03) was dissolved in corn oil (MP Biomedicals, Cat. No. 0290141405) at 30 mg/ml. This was injected intraperitoneally to 6–8 week-old *Srf*[GFAP-CreER]CKO mice at 9 mg/40 g of body weight for 5 consecutive days.

### Immunohistochemistry

Mice were fixed by transcardial perfusion using 4% PFA. The brains were cryoprotected in 30% sucrose, frozen and stored in –80 °C until further use. For staining, 30-µm-thick cryosections were incubated in the blocking/permeabilization solution containing 0.3% Triton-X and 3% goat serum in 1 X PBS (pH 7.4) for 1 hr followed by overnight incubation in the primary antibody. The brain sections were then washed in PBS and incubated in the secondary antibody for 1 hr. The sections were finally mounted in a DAPI-containing mounting medium (Vector Laboratories). The following primary antibodies were used: anti-GFAP (1:1000; Sigma-Aldrich Cat# G3893, RRID:AB_477010), anti-GFAP (1:1000; Agilent Cat# Z0334, RRID:AB_10013382), anti-Sox9 (1:500, R&D Systems, Cat# AF3075, RRID:AB_2194160), anti-Vimentin (1:50; DSHB Cat# 40E-C, RRID:AB_528504), anti-S100β (1:1000; Sigma-Aldrich Cat# S2644, RRID:AB_477501), anti-Iba1 (1:1000; FUJIFILM Wako Shibayagi Cat# 019–19741, RRID:AB_839504), anti-NeuN (1:1000; Sigma-Aldrich Cat# MAB377, RRID:AB_2298772), anti-β-gal (1:1000; Aves Labs Cat# BGL-1040, RRID:AB_2313507), anti-Ki67 (1:500; Vector Laboratories Cat# VP-K451, RRID:AB_2314701), anti-Piccolo (1:400; Synaptic Systems Cat# 142 104, RRID:AB_2619831), anti-GluA1 (1:200; Synaptic Systems Cat# 182 011, RRID:AB_2113443), anti-Tyrosine hydroxylase (1:500, Millipore Cat# AB152, RRID:AB_390204), and anti-MOAB2 (1:500, Novus Cat# NBP2-13075AF532, RRID:AB_2923428). The following secondary antibodies were used: AlexaFluor-488 and –594-conjugated anti-rabbit, anti-chicken, anti-guinea pig and anti-mouse at 1:1000 dilution (Life Technologies). Biotinylated anti-mouse and anti-rabbit secondary antibodies (1:250; Vector Laboratories) were used along with Vectastain ABC kit and ImmPACT VIP kit (Vector Labs). Images were captured using fluorescence microscope (Eclipse 80i, Nikon), or confocal microscope (LSM 880, Zeiss).

### Area measurement of S100β+ve astrocytes

To measure the area of S100*β*+ve astrocytes, images were opened using ImageJ. The background was subtracted from the images using 'Subtract Background' function. The images were then despeckled and binarized. Particles with sizes smaller than 50 µm$^2$ were excluded and the outlines were delineated using 'Analyze Particles' function. The ROIs corresponding to the marked outlines were saved and used to measure the area of the marked cells.

### FluoroJade-C staining

FluoroJade-C (FJC) (#1FJC, Histo-Chem Inc) staining was carried out according to the manufacturer's instructions. Briefly, 35-µm-thick cryosections were mounted on Superfrost plus slides (Brain Research Laboratories) and allowed to dry at 60 °C for 60 min. Slides were immersed in 80% EtOH with 1% NaOH for 5 min, followed by 2 min in 70% EtOH, 2 min in distilled water, and incubated in 0.06% potassium permanganate solution for 10 min. Slides were subsequently rinsed in water, transferred to a 0.0001% solution of FJC in 0.1% acetic acid. The slides were finally rinsed in distilled water, air

dried, cleared in xylene and mounted with DPX mountant. Fluorescence imaging was carried out using an epifluorescence microscope (Eclipse 80i, Nikon) with appropriate excitation/emission filters and captured using Metamorph Software.

## Black-Gold II staining

Black-Gold II staining was done using the Black-Gold II myelin staining kit (AG105, Merck Inc, USA) as per the manufacturer's protocol. Briefly, Black-Gold II powder was resuspended in 0.9% NaCl (Saline solution) to a final concentration of 0.3% and sodium thiosulphate solution was diluted to 1% with MilliQ water freshly before use. 0.3% Black-Gold II and 1% sodium thiosulphate were prewarmed to 60 °C. 35 µm floating brain sections were rehydrated in 1 X PBS for 2 min and then transferred to a pre-warmed Black-Gold II solution. The sections were incubated in Black-Gold II solution at 60 °C for 20 min and then transferred to 1 X PBS and rinsed twice for 2 min each at room temperature. The sections were then incubated with 1% sodium thiosulphate solution at 60 °C for 3 min to stop the reaction. The sections were then rinsed with 1 X PBS thrice for 2 min each at room temperature. The sections were then mounted on positively charged slides and allowed to dry overnight at room temperature. Next day, the slides were dehydrated using a series of graded alcohols, cleared in xylene for 1 min and coverslipped with DPX mounting media. All the Black-Gold II brightfield images were taken on a Nikon Eclipse 80i upright microscope and analyzed using the ImageJ software. For Black-Gold II intensity measurements, the images were converted into 8-bit, binarized and inverted using the invert LUT function in ImageJ. Following this, intensity measurements were made with ROIs of suitable sizes (200x200 µm$^2$).

## Quantification of fluorescence intensity and cell numbers

For measuring fluorescence intensity, images were scaled for 10 X magnification. Ten to 12 areas of field in the same rostro-caudal axis were chosen per image, and the intensities were measured using ImageJ after subtracting the background fluorescence from both control and knockout sections. For the hippocampus, cell numbers or fluorescence intensities were measured in the *stratum oriens* and *stratum radiatum*. For cell counts, images were taken at ×10 magnification. Four ROIs in the same rostro-caudal axis were chosen per image, and the number of cells per ROI were counted with the cell counter plugin using ImageJ and area was converted to mm$^2$.

## Sholl analysis

To measure the hypertrophy of reactive astrocytes, semi-automated Sholl analysis plugged in ImageJ was used. First, the astrocytes and processes of interest were outlined to exclude adjacent cells or areas. Templates of concentric circles from 10 to 100 µm (increasing by 10 µm) were overlaid from the center of the cell soma. For each cell, densitometric thresholds were set to subtract the background, followed by converting to mask, Despeckle and skeletonize the image to analyze the particles. Measurements obtained from each individual cell were recorded. Two-way ANOVA with Sidak's *post hoc* test and mean difference was calculated.

## BBB permeability assay

Assay to measure the integrity of the BBB was performed as previously described (*Andreone et al., 2017*). Briefly, 17-mon old *Srf*$^{GFAP-ER}$CKO (15-mon post-Tam injection) mice were deeply anesthetized with isoflurane and injected with 20 µl of 10 kDa dextran fluorescein (Invitrogen; #D1820) into the left cardiac ventricle, and allowed to circulate for 5 min. Their brains were collected and post-fixed in 4% PFA overnight, frozen and stored at –80 °C. Thirty-µm-thick cryosections were mounted using mounting media supplemented with DAPI (Vector Laboratories Cat# H-1200, RRID:AB_2336790) and analyzed using an epifluorescence microscope (Eclipse 80i, Nikon). Brain sections from the similar rostro caudal (Bregma) position were analyzed. At least 12 different regions were taken and the ratio of the fluorescence intensity (inside versus outside the vessel) was measured.

## Standard qRT-PCR

Quantitative RT-PCR was done with 100 ng of cDNA and KAPA SYBR FAST ABI prism kit (Cat. No. KK4604) using the following program: 95 °C or 3 min followed by 39 cycles of 95 °C for 5 s, 55 °C for 30 s, and 72 °C for 40 s. The PCR reaction was carried out in QuantStudio 7 Flex Real-Time PCR

System (Invitrogen Biosciences). The results were normalized within samples to GAPDH gene expression. All the primer sequences were published previously (*Liddelow et al., 2017*).

## Quantification of the number of synapses

Neocortical sections (30 µm) from tamoxifen injected *Srf*[GFAP-ER]CKO mice and control littermates were immunostained for Piccolo (presynaptic marker) and GluA1 (postsynaptic marker). The images were captured using a confocal microscope (Zeiss LSM880 AxioObserver; PlanApo 63 X/1.4 oil objective) with a lateral sampling of 40 nm and an axial sampling of 1 µm. The area of imaging was optimized as 1024x1,024 pixels and the illumination and imaging conditions (10% power of the lasers, sampling, and pinhole size of 1 Airy unit) were kept constant between the samples. The samples were exported to MetaMorph image analysis module (Molecular Devices, USA) for further segmentation and analysis. The imported multi-wavelength image stacks in 3D were separated into individual channels and a maximum intensity projection was made for each wavelength channel labeling for pre- and postsynaptic markers. The average intensity and standard deviation of each image was obtained through an inbuilt module to quantify image statistics in MetaMorph. The threshold of the images was set at an intensity of (average + standard deviation) for each image. The regions which were between 0.2 µm and 2 µm were automatically detected and filtered to create a binary mask for each image. Clusters with total area less than that of 25 pixel$^2$ (1 pixel = 40 nm) with 0.2 µm (length and breadth) were discarded. The filtering parameters of the thresholded image were chosen to match with the existing information on the size of pre- and postsynaptic compartments obtained through electron microscopy (*Ippolito and Eroglu, 2010*). The binarized images of presynaptic and postsynaptic markers were then analyzed for an overlap for 1 or more pixels between them using the integrated morphometry analysis module in MetaMorph. An overlap of one or more pixels was identified to indicate the presence of a structural synapse. The number of overlapping puncta was counted for each overlaid binary image of pre- and postsynaptic markers. The area of the field for counting synapses was 100x100 µm$^2$.

## Quantification of β-amyloid plaques

The stained brightfield sections were imaged in a Nikon 80i upright microscope. The data were collected as 16-bit images and saved as TIFF format using Metamorph (Molecular Devices). The 16-bit color images were converted into inverted 16-bit monochrome TIFF files and subsequentially used for automated estimation of β-amyloid plaques. The data obtained were pseudo-labelled and blind analyses were performed by a person different from the one who imaged the samples. For every brain section, cortex and hippocampus were marked with randomly chosen regions of interest (ROIs). The ROIs were marked such that in each hemisphere cortex had 2 ROIs of 500x500 pixel$^2$ and hippocampus had three randomly chosen regions 200x200 pixel$^2$ in the *stratum radiatum*. The images were then auto thresholded for dark objects. The Average and Gray level value of thresholded regions were estimated and half this Average Gray level value obtained after initial thresholding was taken as the cut-off intensity for detection of β-amyloid plaques. The thresholded objects were measured using integrated morphometry analysis and objects larger than a pixel area of 100 pixel$^2$ were automatically chosen and counted in each region of interest. The number of plaques in the cortical and hippocampal regions in one brain section were averaged to obtain a mean distribution of plaques across all regions per sample.

## Preparation of hippocampal slices

Acute brain slices were prepared from tamoxifen-treated male and female *Srf*[f/f] (control) and *Srf*[GFAP-ER]CKO mice at 3-mon and 15-mon post-tam administration. Mice were sacrificed by cervical dislocation, and brains were isolated and kept in ice-cold sucrose-based artificial cerebrospinal fluid (aCSF; cutting solution) comprising (mM): 189 Sucrose (Sigma #S9378), 10 D-Glucose (Sigma #G8270), 26 NaHCO$_3$ (Sigma #5761), 3 KCl (Sigma #P5405), 10 MgSO$_4$.7H$_2$O (Sigma #M2773), 1.25 NaH$_2$PO$_4$ (Sigma #8282) and 0.1 CaCl$_2$ (Sigma #21115). After 30 s, the brain was glued to the brain holder of the vibratome (Leica #VT1200), 350-µm-thick horizontal slices were prepared, and the cortex was dissected to isolate the hippocampus. All the slices were kept in slice chamber containing aCSF comprising (mM): 124 NaCl (Sigma #6191), 3 KCl (Sigma #P5405), 1 MgSO$_4$.7H$_2$O (Sigma #M2773), 1.25 NaH$_2$PO$_4$ (Sigma #8282), 10 D-Glucose (Sigma #G8270), 24 NaHCO$_3$ (Sigma #5761), and 2 CaCl$_2$ (Sigma #21115) in water bath (Thermo Fisher Scientific #2842) at 37 °C for 40–45 min. Following recovery, slices were

maintained at room temperature (RT) until transferred to a submerged chamber of ~1.5 ml volume in which the slices were perfused continuously (2–3 ml/min) with warmed (34 °C) and carbogenated aCSF. Post dissection, every step was carried out in the presence of constant bubbling with carbogen (2–5% $CO_2$ and 95% $O_2$; Chemix). All measurements were performed by an experimenter blind to the experimental conditions.

## Extracellular field recordings

Field excitatory postsynaptic potentials (fEPSP) were elicited from pyramidal cells of the CA1 region of *stratum radiatum* by placing concentric bipolar stimulating electrode (CBARC75, FHC, USA) connected to a constant current isolator stimulator unit (Digitimer, UK) at CA3 of Schaffer-collateral commissural pathway and recorded from *stratum radiatum* of CA1 area of the hippocampus, with 3–5 MΩ resistance glass pipette (ID: 0.69 mm, OD: 1.2 mm, Harvard Apparatus) filled with aCSF. Signals were amplified using an Axon Multiclamp 700B amplifier (Molecular Devices), digitized using an Axon Digidata 1440 A (Molecular Devices), and stored on a computer using pClamp10.7 software (Molecular Devices). Stimulation frequency was set at 0.05 Hz. I/O were obtained by setting a stimulation range of 20–40 µs and by adjusting the stimulus intensity by 10 µA per sweep with increments from 0 to 300 µA. Paired pulse ratio (PPR), which is a cellular correlate of release probability of neurotransmitters, was assessed with a succession of paired pulses separated by intervals of quarter log units, with the interval ranging from 3 to 1000ms and were delivered every 20 s. The degree of facilitation (CA3-CA1) was determined by taking the ratio of the initial slope of the second fEPSP relative to the first fEPSP. PPR >1 was considered as a paired pulse facilitation.

Long-term potentiation (LTP) was induced using a theta burst protocol (TBS). A baseline period of 15 min basal fEPSP was recorded at a stimulation intensity that elicited an approximately half-maximal response. Stimulation intensity remained constant throughout the experiment, including during TBS and 60 min post potentiation. Following the baseline period, the TBS protocol was delivered consisting of five bursts (10 stimuli at 100 Hz) at 5 Hz (theta frequency), repeated four times at an interval of 20 s (*Booth et al., 2014*). fEPSP after TBS were recorded for 60 min. Slices with high FV/fEPSP ratio, and unstable for 15 min baseline were discarded from the analysis.

Data analyses were performed using Clampfit 10.7 and Excel 2016. Data are represented as Mean ± SEM. Student unpaired *t*-test were performed to determine the statistical significance. To determine statistical difference in I/O, linear regression was performed on the individual slices to determine the slope of the relationship, and then one-way *ANOVA* (genotype) was performed on the regression points (*Zaman et al., 2000*). Example traces are those recorded for 1–2 min around the time point indicated in the graph.

## Kainic acid (KA) injection

Intracerebroventricular (ICV) injection of KA into the brain was performed as described earlier (*Jin et al., 2009*). Briefly, male, and female *Srf*[GFAP-ER]CKO (4 months post-Tam injection) and control littermates were deeply anaesthetized with 3% isoflurane (Sosrane, NEON, Mumbai, India) for 15 min. The anesthetized animal was placed on a stereotaxic apparatus (KOPF, TUNJUNG, CA, USA) with continuous administration of 1.5–2% isoflurane for 1 hr with vehicle air. Mice were injected with 4 µl saline containing 0.1–0.2 µg of KA using a 10 µl Hamilton syringe fitted with a 28-gauge needle (Hamilton Company, Nevada, USA). The needle was inserted through a hole perforated on the skull (using a dental drill), into the right lateral ventricle using the following coordinates (in mm with reference to bregma): anteroposterior (AP), –0.2; mediolateral (ML), –2.9; dorsoventral (DV), –3.7. After 5 min, the needle was withdrawn over 3 min to prevent backflow. The mice were warmed under infrared (IR) lamp (245 V, 150 W) placed 1–2 feet away until being awakened. After KA injection, animals were carefully monitored for 2 hr and seizure activity was scored by the following Racine score: (0) normal activity, (+1) rigid posture/ immobility, (+2) head bobbing, (+3) forelimb clonus and rearing, (+4) rearing and falling, (+5) tonic-clonic seizures, and (+6) death within 2 hr. The animals were monitored for 7 days and then transcardially perfused. The brains were isolated, post-fixed overnight in 4% PFA, cryoprotected in 30% sucrose, frozen and sectioned at 30 µm in a cryostat (Leica, CM 1850).

## TUNEL assay

The TUNEL assay was carried out using Click-IT Plus TUNEL assay kit (Molecular Probes, Thermo Fisher Scientific). Briefly, 30 μm paraformaldehyde-fixed cryosections were permeabilized with proteinase K solution for 15 min and then incubated with TdT reaction mixture for 60 min at 37 °C and subsequently with FITC-labeled dUTP for 30 min. The slides were washed with 3% BSA in PBS for 5 min and rinsed in 1 X PBS. The slides were mounted using a mounting medium containing DAPI (Vector Labs) and observed using an epifluorescence microscope (Eclipse 80i, Nikon) using appropriate filters and captured using Metamorph software. Numbers of TUNEL-positive cells in the CA1, CA3 and DG regions of entire rostral to caudal brain regions were counted using Image J software. The area of the field for counting the number of TUNEL + cells was 250x250 μm$^2$ and converted to mm$^2$.

## 6-Hydroxydopamine (6-OHDA) injections

6-OHDA (HelloBio, UK, #HB1889) injections in mice were performed as described earlier (*Grealish et al., 2010*). Briefly, the mice, *Srf*$^{GFAP-ER}$CKO (9 mpi) and control littermates were anesthetized with continuous administration of 1.5–2% isofluorane (Sosrane, NEON, Mumbai, India) and placed on the stereotaxic apparatus (KOPF, TUNJUNG, CA, USA). 6-OHDA injections were made in the substantia nigra pars compacta (SNpc) using 10 μl Hamilton syringe fitted with a 28-gauge needle (Hamilton Company, Nevada, USA). The 6-OHDA toxin used was at a concentration of 1.6 μg/μl prepared in ascorbic acid solution (0.2 mg/ml of ascorbic acid in 0.9% saline, filter sterilized). From this, each mouse received a total volume of 1.5 μl using the following stereotaxic coordinates: anteroposterior (AP)=3.0, Mediolateral (ML)=1.2, and dorsoventral (DV)=4.5, with a flat skull position (coordinates in mm with reference to Bregma). Injections were made at 0.5 μl/min with an additional 5 min to allow the toxin to diffuse and 3 min for slow withdrawal of the syringe. Following injections, the mice were warmed under an infrared (IR) lamp (245 V, 150 W) placed 1–2 feet away until being awakened and returned to the home cage. The animals were monitored for 10 days and then transcardially perfused. The brains were isolated, post-fixed overnight in 4% PFA, cryoprotected in 30% sucrose, frozen and 30 μm sections were prepared in a cryostat.

## Open field test

Open field test was used to assess baseline locomotion. The open field test was done under ambient lighting in a 45 (L) x 45 (W) x 35 (H) cm square acrylic box. Mice were gently introduced in the center of the arena and were allowed to explore freely for 10 min. The behavior was recorded using an overhead webcam (Logitech C270HD) at 720 p and 15 fps. Between each test, the arena was thoroughly cleaned with 70% ethanol to remove any odor cues. Movement of mice was tracked using custom MATLAB code developed in our lab (https://github.com/swanandlab). Total distance traveled (cm) was automatically measured using the obtained coordinates. Only male mice (n=11, control; n=9, mutant) were used for all behaviour experiments.

## Fear conditioning

Declarative memory was tested using the contextual fear conditioning paradigm. The mice were handled for 4 days prior to training to reduce any anxiety-related effect on their memory performance. Two days prior to training, mice were habituated to transportation and room environments for 10 min. Configuration of sound attenuated contexts were as follows: context A – floor made up of steel grids, white light, 15% acetone; context B – triangle shed, laminated white sheet as floor to provide different tactile cues,75% ethanol.

### Training

On training days, fear conditioning chambers were cleaned with 15% acetone. The tray below was cleaned and sprayed before placing any animal in the chamber. Animals were allowed to explore context A for 90 s and then a 2.8 kHz 75 db tone was played for 30 seconds. Animals received a 0.7mA mild foot-shock in the last 2 s of 30 s 2.8 kHz 75 dB tone in context A. This was repeated twice with a time interval of 15 s. After the last shock, animals were taken out of the chamber and brought back to their home cage.

### Recent memory recall test in context A
24 hr post-training, animals were scored for their freezing response as a parameter of their memory. The first 90 s of freezing responses were scored manually with a custom scoring assistant plugin in ImageJ. A minimum of 1.2 s continuous freezing bouts were taken as a freezing response. Manually recorded data was analyzed using MATLAB.

### Recent memory recall test in context B
48 hr post-training, animals were scored in context B for their freezing responses for the first 90 s.

### Remote memory recall
One month after the fear conditioning animals were tested again in context B, as explained above, and scored for their freezing response. Twenty-four hr later, the animals were tested in context A and freezing responses were recorded. In the remote recall, the order of contexts was inverted to avoid the CS-US dissociation.

## Barnes maze
Barnes maze test was performed as previously described (*Martyn et al., 2012*). A white circular platform, 92 cm in diameter, with 20 equally spaced 5 cm-wide holes along the periphery was used. The platform was elevated 95 cm from the floor. The behavior was recorded using an overhead webcam (Logitech C270HD) at 720 p and 15 fps. A dark escape chamber was placed under one of the holes, called the target hole, which was indistinguishable from other holes. Three different visual cues, triangular, square, and circular in shape, were placed 10 cm from the edge of the platform, 120° apart from each other for mice to orient themselves in space. Cues remained constant throughout training and testing. Uniform illumination of the platform by bright LED light created an aversive stimulus to motivate the mice to seek the target hole. Before every trial, the arena was cleaned with 70% ethanol to eliminate olfactory cues. In case of unsuccessful trials, mice were gently guided to the target hole. One day before the training trial, the mice were habituated to the maze by placing them in a clear transparent plastic cylinder (15 cm in diameter) in the middle of the platform for 30 s, following which they were guided to the target hole. Mice were given 3 min to enter the escape chamber. The mice that did not enter the escape chamber were gently nudged with the cylinder to enter the chamber. Mice spent 1 min in the escape chamber before being returned to their home cage. Habituation consisted of two trials spaced 4 hr apart. On training day, mice were placed in an opaque plastic cylinder covered by an opaque lid (15 cm in diameter) for 15 s to randomize starting orientation. Each trial was initiated by lifting the cylinder and the trial lasted 2 min. The trial was stopped if the mouse entered the escape chamber. If the mouse did not get into the escape chamber within 2 min of trial, it was gently guided to the target hole with the clear plastic cylinder. Mice were returned to their home cage after 15 s of entering the escape chamber. Training trials were performed twice daily, with 1 hr interval between them with a pseudo-random starting location in the four quadrants. Trials were conducted for 4 consecutive days, until the learning curve leveled off. The test was conducted 2 hr after the last training trial. The escape chamber was removed for the probe test and the mice were allowed to explore the maze for 1 min. The videos were later analyzed for trajectories and other parameters using custom MATLAB scripts (https://github.com/swanandlab; *Nandi, 2023*). Time to reach the correct location of the target hole was reported as latency to the target hole when the body centroid came within 10 cm of the target hole. Time spent in the target quadrant was also measured and reported.

## Astrocyte isolation for transcriptome analysis
Astrocytes were isolated by magnetic assisted cell sorting (MACS) approach using the anti-ACSA2 microbeads (Miltenyi Biotec Inc, Germany). Briefly, the brains were isolated from 5 to 6-week-old $Srf^{GFAP}$CKO mice and meninges were removed in ice-cold homogenization buffer (For 10 ml: 1 ml 10 X HBSS w/o $Ca^{2+}$ and $Mg^{2+}$, 1 µl of 1 mg/ml DNase I, 150 µl of 1 M HEPES, 120 µl of D-glucose, 9 ml milli-Q water). The neocortex and hippocampus were isolated and homogenized in a glass potter with pestle (7 ml volume) in 5 ml of ice-cold homogenization buffer. The homogenate was strained through a 70 µm cell strainer and centrifuged at 300 x *g* for 10 min at 4 °C. The cell pellet was resuspended in 37% Percoll in 15 ml centrifugation tube and centrifuged at 800 x *g* for 20 min at 4 °C. To

make 37% Percoll, first a 100% Percoll solution was prepared by mixing 9 parts of Percoll with 1 part of 10 X DEPC-treated PBS (DPBS). This 100% Percoll was diluted to 37% using 1 X DPBS. The myelin in the top layer was carefully removed and discarded. The remaining supernatant was transferred to a fresh 50 ml centrifugation tube and 3 volumes ice-cold PB buffer (0.5% BSA in 1 X DPBS) to dilute the Percoll. The cell pellet was gently resuspended in 5 ml ice-cold PB buffer. Both these tubes were centrifuged at 700 x $g$ for 10 min at 4 °C. The supernatant was discarded, and the cell pellets were pooled together in 80 µl PB buffer and 10 µl of FcR blocking reagent was added per $10^7$ cells. After 10 min incubation at 4 °C, 10 µl of anti-ACSA2 microbeads were added per $10^7$ cells and incubated for 15 min at 4 °C. The ACSA2 +astrocytes were isolated using the midi-MACS separator and LS columns (Miltenyi Biotec) as per the manufacturer's protocol. The ACSA2+ astrocytes were eluted from the magnetic columns and pelleted by centrifugation at 300 x $g$ for 10 min at 4 °C and resuspended in RNA Later (Invitrogen). To measure the purity of isolated astrocytes, the eluted ACSA2+ cell fraction was plated on poly-D-lysine coated cover glass in 24-well dishes and grown in DMEM medium containing 10% FBS (Invitrogen) and 1% penicillin/streptomycin (Invitrogen). Cells were fixed at 7 days in vitro and immunostained for GFAP. The purity of the culture was around 95% (based on GFAP+/ total DAPI+ cells).

## RNA sequencing and analysis

The sequence data was generated using Illumina HiSeq 2500. Data quality was checked using FastQC and MultiQC software. The data was checked for base call quality distribution, % bases above Q20, Q30, %GC, and sequencing adapter contamination. All the samples passed QC threshold of $Q_{20} >$ 95%. Raw sequence reads were processed to remove adapter sequences and low-quality bases using fastp. The QC passed reads were mapped onto indexed Mouse reference genome (GRCm 38.90) using STAR v2 aligner. On average 99.03% of the reads aligned onto the reference genome. Gene level expression values were obtained as read counts using featureCounts software. Expression similarity between biological replicates was checked by spearman correlation and Principal Components Analysis. For differential expression analysis the biological replicates were grouped as Control and Test. Differential expression analysis was carried out using the edgeR package after normalizing the data based on trimmed mean of M (TMM) values. After normalization 27540 features (52.32%) have been removed from the analysis because they did not have at least 1 counts-per-million in at least four samples. Genes with absolute $\log_2$ fold change ≥ 1.5 and adjusted p-value ≤ 0.05 were considered significant. The expression profile of differentially expressed genes across the samples is presented in volcano plots and heatmaps. The genes that showed significant differential expression were used for Gene Ontology (GO) and pathway enrichment analysis using DAVID (https://david.ncifcrf.gov/).

## Experimental design and statistical analysis

Prior to starting each experiment, animals were randomly assigned to experimental groups. For phenotypic studies, experiments were repeated in separate cohorts. The experimenter was blinded to the genotype of the mice in all surgical, histological, electrophysiological, and behavioral analyses. The comparisons between two groups were done using unpaired two-tailed Student's $t$-test or one-way ANOVA with Tukey post hoc test. For our group comparisons with two variables (context and genotype) in the fear conditioning test, two-way ANOVA was used followed by Sidak's post hoc test to analyze the interactions between the wo variables. The standard error mean (SEM) was calculated and depicted as error bars in the graphs. All the statistical details for each experiment, including the $n$ value, the statistical test used, p value, significance of comparisons are mentioned in the figure legends. Analyses were done using GraphPad Prism 6 (Graphpad Software Inc; LaJolla, CA; RRID:SCR_002798). A p<0.05 was considered significant in all statistical analyses. No samples were omitted from the analysis. Data presented in the study represent biological (and not technical) replicates. Both male and female mice are included in all experiments except the behavioral experiments, in which only male mice were used.

## Acknowledgements

We thank Dr. Suzanne Baker (St. Jude's Hospital, Memphis, TN, USA) for generously sharing the GFAP-ERT transgenic mouse line; Dr. Joseph LoTurco (Univ. of Connecticut, Storrs, CT, USA) for sharing the piggyBac plasmids. Dr. Hiyaa Ghosh (NCBS, Bangalore) for sharing antibodies; We thank the IISc

Bioimaging Facility and the Central Animal Facility for confocal imaging and animal care, respectively. SwarnaJayanti Fellowship, Department of Science and Technology, India DST/SJF/LSA-01/2012–2013 (NR) Science and Engineering Research Board grant CRG/2019/006899 (NR) Department of Biotechnology (DBT)-IISc Partnership Program grant (NR, DN) Science and Engineering Research Board grant EMR/2015/001946 (JPC) INSPIRE Faculty Fellowship DST/INSPIRE/04-I/2016–000002 (SM) National post-doctoral fellowship PDF/2017/001385 (SCRT) Senior research fellowship, University Grants Commission, India (MJ, SD) Senior research fellowship, Council for Scientific and Industrial Research, India (AN)

## Additional information

### Funding

| Funder | Grant reference number | Author |
| --- | --- | --- |
| Department of Science and Technology, Ministry of Science and Technology, India | DST/SJF/LSA-01/2012-2013 | Narendrakumar Ramanan |
| Science and Engineering Research Board | CRG/2019/006899 | Narendrakumar Ramanan |
| Department of Biotechnology, Ministry of Science and Technology, India | BT/PR27952/INF/22/212/2018 | Deepak Nair |
| Science and Engineering Research Board | EMR/2015/001946 | James P Clement |
| Department of Science and Technology, Ministry of Science and Technology, India | DST/INSPIRE/04-I/2016-000002 | Swananda Marathe |
| Science and Engineering Research Board | PDF/2017/001385 | Surya Chandra Rao Thumu |
| University Grants Commission | | Monika Jain Soumen Das |
| Council for Scientific and Industrial Research (CSIR), India | | Arnab Nandi |

The funders had no role in study design, data collection and interpretation, or the decision to submit the work for publication.

### Author contributions

Surya Chandra Rao Thumu, Monika Jain, Soumen Das, Formal analysis, Validation, Investigation, Writing – review and editing; Sumitha Soman, Vijaya Verma, Arnab Nandi, Formal analysis, Investigation, Writing – review and editing; David H Gutmann, Resources, Writing – review and editing; Balaji Jayaprakash, James P Clement, Swananda Marathe, Resources, Formal analysis, Writing – review and editing; Deepak Nair, Resources, Formal analysis, Investigation, Writing – review and editing; Narendrakumar Ramanan, Conceptualization, Supervision, Funding acquisition, Investigation, Writing - original draft, Project administration, Writing – review and editing

### Author ORCIDs

Surya Chandra Rao Thumu http://orcid.org/0000-0002-2089-9792
Soumen Das http://orcid.org/0000-0001-6422-0238
David H Gutmann https://orcid.org/0000-0002-3127-5045
Balaji Jayaprakash http://orcid.org/0000-0002-4442-6981
Swananda Marathe https://orcid.org/0000-0002-2539-366X
Narendrakumar Ramanan http://orcid.org/0000-0002-6088-9599

## Ethics

All the procedures in this study were performed according to the rules and guidelines of the Committee for the Purpose of Control and Supervision of Experimental Animals (CPCSEA), India. The research protocol was approved by the Institutional Animal Ethics Committee (IAEC) of the Indian Institute of Science (Protocol numbers: CAF/Ethics/596/2018 and CAF/Ethics/731/2020).

## Decision letter and Author response

Decision letter https://doi.org/10.7554/eLife.95577.sa1
Author response https://doi.org/10.7554/eLife.95577.sa2

---

# Additional files

## Supplementary files

• MDAR checklist

## Data availability

All data generated or analysed during this study are included in the manuscript and supporting file.

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
