## [Editor Report]

This is an important study that uses a variety of complementary approaches to demonstrate that adult astrocytes lacking serum response factor are neuroprotective. The evidence supporting this conclusion is solid, reflecting mostly high quality cellular and molecular data with minor remaining concerns regarding the behavioral data.

---

## [Decision Letter]

[Editors' note: this paper was reviewed by Review Commons.]

---

## [Author Response]

General Statements [optional]

We thank the editors for sending our manuscript for peer review and the reviewers for careful reading and their critical comments to improve the manuscript. Below, we describe the experiments that have been carried out in response to the reviewers and incorporated in the preliminary revision. We have addressed all concerns of the reviewers, and the experiments suggested have helped to strengthen the study reported in this manuscript without affecting the fundamental findings.

Point-by-point description of the revisionsReviewer #1 (Evidence, reproducibility and clarity (Required)):­Summary: The authors used a conditional transgenic mouse model to demonstrate that deletion of serum response factor (SRF) from adult astrocytes provides neuroprotection in various insult/ diseases contexts without promoting any obvious phenotypic deficiencies. The work builds on the group’s previous study where SRF was embryonically deleted from astrocytes and their precursor cells. Given the role of SRF in promoting glial cell differentiation, the adult conditional KO used in the current study was designed to circumvent the limitations of the previous approach. The authors used a variety of complementary approaches (including immunohistochemistry, electrophysiology, transcriptomics, and behavior) to demonstrate the therapeutic potential of their approach. However, I have questions regarding the validity of the behavioral analyses as well as some of the imaging results that dampen my overall enthusiasm.Major Comment #1The synaptogenic factors probed in Figure 3C (e.g. glypicans, thrombospondins, etc.) are not likely to play major roles in the adult brain in a non-injury context, so I do not know that these analyses provide any significant insight into potential functional changes in the mutant mice. Along the same lines, the analysis of synapse count (Figure 3D-E) seems inconsequential given that SRF was knocked out well after the period of developmental synaptogenesis. It would have been much more interesting to have performed these analyses following insult (such as the kainate injury model used by the authors) or in one of the disease models presented later in the manuscript. As it stands, I don't think they add very much to the study.

We are grateful to the reviewer for the careful reading of the manuscript. Astrocytes are known to regulate the formation, maintenance, and elimination of synapses. It has been previously shown that LPS-induced reactive astrocytes exhibit reduced expression of several synaptogenic factors, were unable to promote synapse formation and showed reduced phagocytic activity (PMID: 28099414). We wanted to determine whether the SRF-deficient reactive-like astrocytes were likely compromised in their ability to produce pro-synaptogenic factors and/or adversely affect synapse maintenance. We agree with the reviewer that analysis of synapses in the adult brain may not address the role of these mutant astrocytes in synaptogenesis. But our results indicate that the mutant astrocytes are likely not affecting synapse maintenance or exhibit altered phagocytotic activity that would result in increased or decreased synapse numbers. We have made this clear in the revised manuscript.

Major Comment #2There is considerable variability in the behavioral results, particularly the fear conditioning and Barnes maze tasks (Figures 4F-G). Given the extremely low sample size for mouse behavior (n=5 in on group, n=7 in the other), it is highly likely that the behavioral tests were done with a single cohort of animals (which would be far from ideal) and that these experiments are significantly underpowered. Furthermore, it does not appear that the fear conditioning task was properly optimized. For example, in the control mice in context A, there were two animals that were at or very close to 0 percent freezing; these were likely outliers, or even an indication that the foot shock conditioning protocol was not working as it should. The highest percent freezing of either group was ~70%, which would have been an ideal starting place as an average for the control group. In addition, sex of the animals was not reported for these experiments. If the authors combined sexes as they did in other analyses in this paper, it is possible that they missed reaching the appropriate reaction threshold for the foot shock for some of the animals, as sex differences have previously been demonstrated in mice (DOI: 10.1037/bne0000248). Given the age at which the animals are assessed with these tasks, these specific revisions would require greater than 6 months to complete. However, as currently presented, there simply are not enough data points to make any conclusions regarding behavior.

We have performed the behavioural experiments with an additional cohort of animals for both control and mutant groups and reanalysed the data. We now have n=11 for control and n=9 for mutant group. Only males were used for the behaviour experiments, and we have indicated this in the revised Methods section (page 31). We do not see any significant difference in behaviour between the two groups. These results are included in revised Figure 4E-G in the revised manuscript.

Minor Comment #1:The representative GFAP images (Figure 1 E/G) do not appear to have been taken at the same magnification. This was particularly apparent in the comparison between the control and CKO hippocampus at 12mpi. It is difficult to say with certainty, due to the lack of fiducial markers in many of the images. Inclusion of a nuclear stain (DAPI) would be highly beneficial to allow the reader to make a more informed comparison.

These images were taken at the same magnification. We observed that the mutant astrocytes are much larger in older mice (12 months post-Tam injection) compared to younger mice (2 months post-Tam injection). This is one of the possilbe reasons that the magnification may appear different. We have included the DAPI staining for these images in Figure 1—figure supplement 2 in the revised manuscript.

Minor Comment #2:The authors should note that the use of GluA1 as a postsynaptic marker will not identify silent synapses (i.e. structurally "normal" but functionally inert).

We agree with the reviewer that GluA1 will not identify functionally silent synapses. We have indicated that we are only looking at structural synapses in the revised manuscript.

Referees cross-commentingAfter reading the comments of the other reviewer, I think we're in agreement that the cellular and molecular data, while descriptive, is of mostly excellent quality. Moreover, the significance of the study is high, and the potential readership broad. However, I stand by my initial assessment of the behavioral data and find the manuscript quite lacking in this regard. Proper revisions would take at least half a year or more, so the authors may be disinclined to go this route. That being said, if the behavioral data were to be excised, I would be happy to sign off on the rest of the manuscript provided that the other major criticisms are addressed.

We thank the reviewer for the appreciation of our work. We have increased the number of animals in the behavioural experiments and do not see any significant difference between the two groups. These results are included in revised Figure 4E-G in the manuscript.

In response cross-comment of Rev 2:Agreed that if properly conducted and presented, the behavioral data would indeed provide a nice functional correlate to the cellular work. In its current state, I'm afraid that it is instead a hindrance to the study and I would recommend that they just remove it if they choose not to address my concerns with the quality (particularly the extreme variability and the complete lack of freezing by several of the animals, especially in the controls).

We hope that the revised behaviour data would provide a strong functional correlate to the other findings in the study.

Additional cross-comments:I agree with the added criticisms raised by Reviewer #3, and I think that the manuscript would be greatly improved by revisions that address those and the original criticisms from myself and Reviewer #2. I still think that the behavioral data should be omitted, provided that the authors are not capable or willing to appropriately address those concerns within a reasonable time frame.

We have addressed all the concerns raised by Reviewer 3.

Reviewer #2 (Significance (Required)):The manuscript addresses the important area of the cellular mechanisms that underlie neuroprotection. The ms adds to our understanding of genetic control of neuroprotection and should be of significant interest to others in the field. The experimental approach systematic and the data presented are generally of high quality and believable. While the ms presents quite a bit of overall cellular data that underlies various areas of neuronal and brain function that may be affected by loss of SRF, it is still somewhat descriptive. It is unclear what aspect of astrocyte reactivity is determinative, how mechanistically in normal cells SRF suppresses reactivity, and how SRF -negative reactive astrocytes confer such broad neuroprotection. While the latter is well beyond the scope of this study, the authors do propose SRF may be involved in regulating oxidative stress and amyloid plaque clearance as a potential pathway to account for SRF's role, however a more systematic discussion based on the gene expression data and known pathways would be welcome. Overall, this is a high quality ms that should be of interest to the field that identifies a SRF as a novel player in neuroprotection.1. Quantification of the extent of SRF loss in astrocytes in conditional tamoxifen knockout would strengthen the quality of the data.

We have provided this data in revised Figure 1C.

2. While the authos did use a Sholl analysis to show hypertophic changes in SRF negative astrocytes, given that SRF is an important regulator of actin and other cytoskeletal related proteins in other cell types, and that cytoskeletal components can play an important role in cell signaling, it is somewhat surprising that the gene array analysis did not include actin and other cytoskeletal proteins, nor did the authors consider a more careful analysis of intracellular cytoskeletal changes and the potential mechanistic implications of this for observed reactivity and neuroprotection.

We agree with the reviewer that SRF is a well-established regulator of actin cytoskeleton. However, we did not check for any significant changes in gene expression for actin or actin-regulatory proteins. We would have expected a decrease in astrocyte morphology similar to the neurite/axon defects exhibited by SRF-deficient neurons. It is unclear whether the hypertrophic morphology is due to transcriptional regulation of actin/actin-binding proteins or due to astrocyte reactivity. This would be a very interesting question and we will investigate these aspects in future studies.

Reviewer #3 (Evidence, reproducibility and clarity):Summary: The study by Thumu et al., suggests that astrocytic specific deletion of SRF in mice results in morphological changes in these cells that does not affect neuronal survival, synapse number, plasticity or cognition. However, in in vivo mouse models of excitotoxic damage and neurodegenerative disease, deletion of SRF reduced neurotoxicity. The authors provide sufficient evidence to suggest that astrocytic SRF contributes to neurotoxicity in various models however some claims are made that are currently not supported by evidence.

We thank the reviewer for critical reading of our manuscript and for suggesting ways to strengthen our findings. We have addressed all the concerns raised by the reviewer and with additional experiments where required. We hope that these changes would have improved our manuscript.

Major comments:1) The title of the manuscript is "SRF-deficient astrocytes provide neuroprotection in mouse models of excitotoxicity and neurodegeneration". It would be more accurate to say that SRF is involved in neurotoxicity in these models. To make a comment on the role of SRF in neuroprotection, experiments should be performed in spinal cord injury or ischaemia, where deficiency of SRF would be hypothesised to worsen recovery.

We disagree with the reviewer with this assessment. There is no evidence to suggest that SRF is involved in neurotoxicity. What our data suggests is that SRF deficiency results in a reactive astrocyte state that is neuroprotective in these models. We hypothesize that in injury/infection/disease conditions that would result in generation of neuroprotective astrocytes, SRF expression or function may be negatively regulated. It would be interesting to see whether the SRF-deficient astrocytes alleviate or exacerbate pathology and recovery following spinal cord injury and ischaemia.

2) The authors claim that SRF KO astrocytes undergo hypertrophy. However, the quantification of the number of intersections gives information about morphology rather than hypertrophy. Quantification of cell size (area of S100B staining) should be provided.

We have quantified the cell area based on S100B staining and have included this data in revised Figure 1E.

3) In Figure S1 the authors provide evidence showing lack of B-gal in cell types other than astrocytes (neurons/OPCs). However, microglia are missing, which could be important as later they show that microglia undergo changes in the SRF knockout model. This staining should be provided.

We have performed double immunostaining for B-gal and IbaI and do not see any overlap between these two, suggesting that there is no Cre expression in microglia. We have included this data in Figure 1—figure supplement 1 in revised manuscript and mentioned this in the revised text (page 5).

4) Can the authors explain the large amount of variability in number of synapses in 15 mpi in Figure 3E?

We have quantified synapses from two more mice and added that data, and the variability is less. The revised data is included in revised Figure 3E.

5) The authors claim in the text that microglia have thicker processes and an amoeboid shape however no evidence of this is provided in Figure S5.

We have provided higher magnification images to show amoeboid microglial morphology in revised Figure 3—figure supplement 3.

6) For the RNAseq of isolated astrocytes did the authors confirm that other cell types (e.g microglia) did not contaminate their samples?

To test purity, the ACSA2+ fraction was plated on poly-D-lysine coated coverglass and grown in DMEM containing 10% FBS and 1% antibiotic/antimycotic. Cells were fixed at 7 days later and immunostained for GFAP. We found the purity to be around 95% based on GFAP+/DAPI+ cells. We have provided this information in the revised methods section of the manuscript.

7) In the text "Enrichment analysis of Gene Ontology terms for Biological Process (GO BP) revealed that Srf deficient astrocytes showed enrichment of pathways related to cellular response to β amyloid and β-amyloid clearance." This is not shown in Figure 5. It would be more accurate to say that there is a downregulation of genes involved in B amyloid metabolic process.

We apologize for the omission in showing that this data was presented in the revised Figure 5—figure supplement 1. We now state that, “there is downregulation of genes involved in b-amyloid metabolic process”, as suggested by the reviewer.

Minor comments:1) The authors say that in Figure 1B many astrocytes did not show any SRF expression. However, overall averages of SRF intensity are plotted in Figure 1C. It would support their claim to instead to calculate the percentage of SRF expressing cells above a certain threshold in each condition, rather than plotting the mean intensity. As a control for their method of quantifying SRF intensity in Figure 1B, demonstrating no change in SRF in neurons would provide confidence for the specificity of the knockout.

We have provided the quantification of the extent of SRF loss in astrocytes (percent astrocytes that are deleted for SRF) as suggested by Reviewer 2 in Revised Figure 1C and in page 5 in the main text, as this would provide a more meaningful information.

2) The authors use the term "reactivation" throughout the manuscript. This could be misconstrued as re-activation and so I would suggest using the terms "reactivity" or "reactive transformation". Furthermore, only one region is quantified in Figure 1C while in later figures multiple regions are quantified. The authors should justify this decision or update the figures with data from missing regions.

We have changed “reactivation” to “reactivity” in all places as suggested by the reviewer.

3) In Figure S2 the authors should provide a positive control for their staining.

We have provided positive control for these staining experiments in Figure 2—figure supplement 1.

4) Figure 1E is missing body weight data noted in the figure legend.

We apologize for this oversight. This data was previously included in Figure 3—figure supplement 1 and not in Figure 1. We have made this correction to Figure 1 legend.

5) Images in Figure 2C are poorly visible and should be improved in terms of either quality or magnification.

We apologize for the poor quality of the images in Figure 2A. We have now provided higher magnification of the images in revised Figure 2A.

6) In Figure 2B figure labels are missing.

We thank the reviewer for pointing out this omission. We have added the missing labels.

7) Details of houskeeping gene normalisation are missing from qPCR data.

We apologize for not providing this information. We have included this in the revised Methods section.

8) The authors should provide a list of differentially expressed genes from RNAseq of SRF KO mice. No information is currently given in the text about the number of differentially expressed genes in the conditional knockout.

We have provided a list of differentially expressed genes from RNAseq of SRF KO mice as Figure 5-source data1 and 2. We have also provided information on the number of differentially expressed genes in the revised manuscript.

9) In figure 5A data would be better illustrated as a volcano plot (similar to Figure S7C).

We have replaced the heatmaps with volcano plaots in figure 5A as suggested by the reviewer.